# Beyond Blind Noising: Disentangled Visual Rectification for Hallucination Mitigation in MLLMs

Yujia Chen[1]  Rui Sun[2]  Bingzhou Wang[1]  Huayu Mai[1]  Wangkai Li[1]  Zhaoyang Li[1]  Aibing Li[1]
Wenzhang SUN[3]

## Abstract

Visual Contrastive Decoding (VCD) mitigates hallucinations in Multimodal Large Language Models (MLLMs) by penalizing the output shift from noise-perturbed images, assuming this shift captures the hallucination direction. We prove this assumption flawed: noise-induced drift in Language-Image Pretrained (LIP) encoders is a *coupled vector* entangling (i) structural degradation from corrupted visual information with (ii) hallucination induction from linguistic prior activation. VCD's indiscriminate penalty inevitably suppresses valid visual semantics. Our key insight is that Self-Supervised Learning (SSL) encoders exhibit *only* structural degradation under noise—geometrically orthogonal to hallucination paths—enabling principled disentanglement via LIP–SSL differential response. We propose **Disentangled Visual Rectification (DVR)**, a training-free dual-stream framework performing visual-layer rectification and decoding-layer contrast on purified representations. DVR achieves approximately $5\times$ theoretical error reduction over VCD and establishes SOTA performance on POPE, MME, LLaVA-Bench and CHAIR benchmarks. Code is available in https://github.com/money-abundance-infinite/DVR.

## 1. Introduction

Multimodal Large Language Models (MLLMs) (Liu et al., 2023b;c; Zhu et al., 2023; Bai et al., 2023; Achiam et al.,

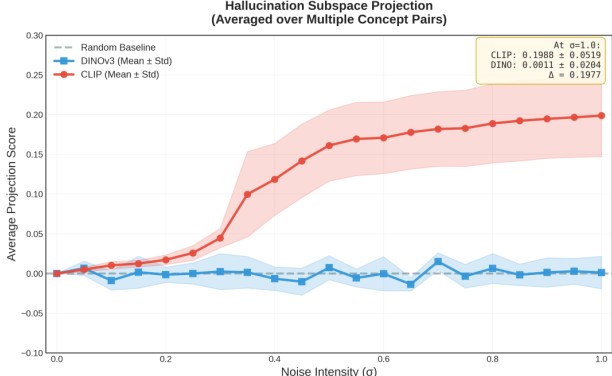

*Figure 1.* **Geometric Analysis of Noise-Induced Feature Drift.** We project the visual feature drift vector $\Delta\mathbf{z}$ onto a pre-defined semantic hallucination direction $\mathbf{v}_h$ (*e.g.*, *Dining Table → Vase*) across varying noise intensities $\sigma$. Shaded regions denote standard deviation across multiple concept pairs. **(Red)** CLIP exhibits significant positive correlation ($\mathcal{S} \approx 0.20$ at $\sigma = 1.0$), confirming that noise triggers directional drift toward linguistic priors (hallucination induction). **(Blue)** In contrast, **DINOv3** maintains near-zero projection ($\mathcal{S} \approx 0.00$), demonstrating that its feature degradation remains orthogonal to semantic hallucination directions, reflecting pure structural information loss. Detailed experimental settings are provided in Appendix A.

2023) have achieved landmark breakthroughs in artificial general intelligence by deeply integrating the powerful reasoning capabilities of Large Language Models with the perceptual abilities of visual encoders. Despite their remarkable performance on visual question answering (Liu et al., 2023c;b; Hudson & Manning, 2019) and complex reasoning tasks (Lu et al., 2022; Alayrac et al., 2022; Achiam et al., 2023), these models remain fundamentally plagued by *hallucination* (Li et al., 2023b; Fu et al., 2023b; Yue et al., 2024)—the generation of textual content that contradicts the input image. Notably, hallucination in MLLMs differs in essence from factual hallucinations in text-only LLMs, which originate from errors in parametric knowledge: **the core issue lies in a systematic failure of cross-modal grounding** (Leng et al., 2024; Wang et al., 2024d). Under this failure mode, models tend to neglect input visual evidence and instead over-rely on linguistic statistical priors acquired from training corpora. For instance, when pro-

[1]School of information Science and Technology, University of Science and Technology of China, China [2]Shenzhen International Graduate School, Tsinghua University, China [3]School of Software, Tsinghua University, China. Correspondence to: Rui Sun <issunrui@mail.ustc.edu.cn>.

*Proceedings of the 43rd International Conference on Machine Learning*, Seoul, South Korea. PMLR 306, 2026. Copyright 2026 by the author(s).

cessing an image of a "living room" scene, a model may fabricate non-existent objects solely based on the high co-occurrence probability of "living room" and "sofa" in text. This systematic deficiency in visual grounding not only reduces models to "blind" language completion machines but also critically impedes their reliable deployment in high-stakes domains such as autonomous driving and medical diagnosis.

Existing approaches to MLLM hallucination mitigation fall into three paradigms: *data-driven fine-tuning* (Liu et al., 2025; Yang et al., 2025c; Yoon et al., 2025; Zhuang et al., 2025), *internal intervention* (Che et al., 2025; Dong et al., 2025; Wang et al., 2024a; Liu et al., 2024b; Ghosh et al., 2024), and *contrastive decoding* (Huang et al., 2025; Leng et al., 2024; Wang et al., 2024d; Wan et al., 2025). Among these, training-free **Visual Contrastive Decoding (VCD)** (Leng et al., 2024) has emerged as the predominant approach owing to its plug-and-play nature, strong generalizability, and interpretability. The core mechanism of VCD involves applying noise perturbations to input images to amplify hallucinations, followed by suppression via logit-level contrastive decoding:$Logits_{\text{VCD}} = Logits_{\text{original}} + \alpha \left( Logits_{\text{original}} - Logits_{\text{noise}} \right)$. This formulation embeds a critical assumption: that the output shift induced by noise perturbation aligns precisely with the "hallucination direction," such that inverse correction achieves effective suppression. **However, we raise a fundamental challenge to the validity of this assumption: does noise-induced variation truly equate to a pure hallucination direction?**

Our theoretical analysis demonstrates that **the answer is negative**. In an ideal hallucination mitigation framework, the correction direction should point toward a hallucination-free "Oracle" state—*i.e.*, the ideal ground-truth output distribution—with the correction vector defined by $Logits_{\text{oracle}} - Logits_{\text{original}}$. VCD attempts to approximate this ideal direction using $\Delta = Logits_{\text{original}} - Logits_{\text{noise}}$; however, this approximation introduces **severe systematic bias**. The fundamental issue lies in the fact that noise-induced feature drift constitutes a **coupled vector** that simultaneously encompasses two fundamentally distinct components: **(1)Structural degradation component**: the loss of genuine visual information caused by noise-corrupted visual structures; **(2)Hallucination-inducing component**: the drift toward erroneous textual concepts triggered when ambiguous features induce the model to rely on linguistic priors.

This coupling effect pervades mainstream MLLM architectures, as their visual encoders (*e.g.*, CLIP (Radford et al., 2021), SigLIP (Zhai et al., 2023)) adhere to the **Language-Image Pre-training (LIP)** paradigm. Even in advanced architectures such as Qwen2.5-VL (Bai et al., 2025b), the optimization objective remains maximizing image-text alignment. Consequently, when visual evidence becomes am-

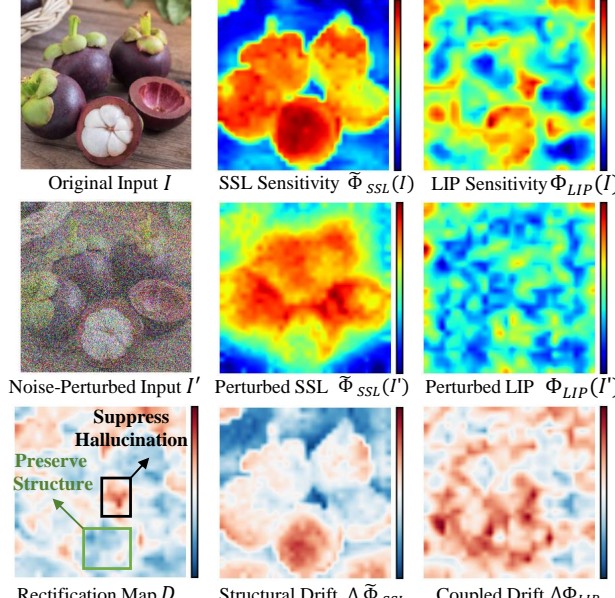

Original Input $I$    SSL Sensitivity $\tilde{\Phi}_{SSL}(I)$    LIP Sensitivity $\Phi_{LIP}(I)$

Noise-Perturbed Input $I'$    Perturbed SSL $\tilde{\Phi}_{SSL}(I')$    Perturbed LIP $\Phi_{LIP}(I')$

Rectification Map $D$    Structural Drift $\Delta\tilde{\Phi}_{SSL}$    Coupled Drift $\Delta\Phi_{LIP}$

*Figure 2.* **Visualizing Paradigm Asymmetry and Disentanglement.** Under noise perturbation ($I \rightarrow I'$), the SSL encoder exhibits pure structural drift ($\Delta\tilde{\Phi}_{\text{SSL}}$), whereas the LIP encoder manifests coupled drift ($\Delta\Phi_{\text{LIP}}$) that conflates structural loss with linguistic prior-induced completion. DVR exploits this asymmetry to derive the Rectification Map ($\mathbf{D}$, right), enabling selective intervention: suppressing hallucination-dominant regions (red, $\mathbf{D} > 0$) to attenuate spurious semantic activation, while preserving structure-loss regions (blue, $\mathbf{D} < 0$) to recover genuine visual details.

biguous due to noise, LIP encoders do not merely lose information—they instinctively retrieve aligned textual concepts to "complete" the representation, thereby activating prior-based hallucination induction.

This directly exposes the theoretical flaw of VCD: the correction vector $\Delta$ is essentially a **coupled superposition** of structural degradation and hallucination induction. Since VCD employs an indiscriminate inverse penalty mechanism, it uniformly suppresses all noise-induced feature variations—both the spurious concept drift targeted for elimination and the genuine visual details lost to perturbation. This "collateral damage" inevitably attenuates noise-sensitive yet semantically valid fine-grained features, causing irreversible degradation of semantic fidelity.

**How can we decouple the pure hallucination direction from structural degradation?** Our key insight emerges from a principled deconstruction of visual encoder training paradigms. LIP encoders, represented by CLIP and SigLIP, operate in a feature space that constitutes a manifold where visual and textual representations are deeply entangled. When visual inputs become ambiguous due to noise, the model activates a **"semantic completion" mech-**

anism grounded in embedded linguistic priors. Owing to the strong coupling between visual and textual features, this completion behavior is not isotropic random walk but rather tends to drift along the **gradient of co-occurrence priors** toward hallucinated concepts with high frequency in the training corpus (*e.g.*, a noisy "dining table" feature does not degrade into meaningless noise but is "completed" toward the frequently co-occurring "vase" text embedding). This directional drift constitutes the explicit **hallucination-inducing component**. **In stark contrast**, pure visual self-supervised learning (SSL) encoders such as DINOv3 (Siméoni et al., 2025) optimize exclusively for intra-image structural discrimination and local-global geometric correspondence, remaining inherently decoupled from textual supervision. This implies that SSL encoders lack the capacity to "fill in" representations using linguistic statistics. Their noise response manifests as pure **visual structural degradation**—a direction that maintains **strict geometric incoherence** with hallucination paths pointing toward erroneous semantic concepts (illustrated in Figure 1).

Building upon this insight, we propose **Disentangled Visual Rectification (DVR)**, a training-free dual-stream framework that exploits the geometric orthogonality between LIP and SSL feature spaces. By computing the differential response between both encoders under noise perturbation, we leverage their complementary properties—cross-modal alignment *vs.* unimodal structure preservation—to cancel the shared structural degradation component, thereby isolating the hallucination-inducing direction. We theoretically prove that this differential operation serves as an effective proxy for subspace projection in high-dimensional spaces and establish explicit upper bounds on residual error (Appendix B).

DVR operates through two synergistic streams that address hallucination at complementary stages of the generation pipeline: **Stream I: Visual-Layer Rectification.** At the visual encoding layer, we leverage the decoupled noise difference map across heterogeneous visual encoders to adaptively modulate the original visual tokens. Specifically, in **hallucination-dominant regions**—where LIP feature variation significantly exceeds SSL variation—we apply suppressive gain to attenuate misleading features that arise from linguistic prior activation. Conversely, in **structural-loss-dominant regions**—where SSL variation is substantial—we perform feature enhancement to compensate for genuine visual details lost to noise perturbation. This selective modulation generates semantically faithful and hallucination-robust visual representations *at the source*, as illustrated in Figure 2. **Stream II: Decoding-Layer Contrast.** While Stream I purifies visual representations before LLM processing, visual token influence attenuates with decoding depth—a "visual fading" phenomenon (Huang et al., 2024; Leng et al., 2024) (further empirical analysis in Ap-

pendix F). This creates opportunities for textual priors to re-dominate generation. We therefore introduce a logit-level contrastive mechanism operating on the purified features. Since inputs have already undergone disentanglement, the contrastive signal remains uncontaminated by structural degradation, enabling higher-precision targeting of residual hallucinations. Theoretical analysis demonstrates that DVR achieves approximately five-fold spurious error reduction compared to VCD.

**Contributions.** (1) We provide the theoretical analysis revealing that noise-induced feature drift conflates structural degradation with hallucination induction, exposing a fundamental flaw in VCD. (2) We propose **DVR**, a training-free dual-stream framework that exploits LIP-SSL geometric orthogonality for principled disentanglement, with provable error reduction guarantees. (3) DVR significantly outperforms SOTA methods on POPE, MME, LLaVA-Bench and CHAIR benchmarks.

## 2. Related Work

### 2.1. Mitigating Hallucinations in MLLMs.

Recent efforts to mitigate object hallucinations in MLLMs generally fall into three paradigms: data optimization, attention intervention, and contrastive decoding. Data-centric approaches align models via negative or counterfactual data construction (Liu et al., 2023a; Yu et al., 2024; Wang et al., 2024d) and refine generation through contrastive learning (Sarkar et al., 2024; Jiang et al., 2024a; Chen et al., 2025; Shang et al., 2024) or reinforcement learning strategies (Liu et al., 2025; Lyu et al., 2024; Suo et al., 2025; Yang et al., 2025c; Yoon et al., 2025; Zhuang et al., 2025). Alternatively, internal intervention methods operate during inference by manipulating feature spaces or attention mechanisms, employing techniques such as feature enrichment (Wang et al., 2025; Duan et al., 2025; Che et al., 2025; Dong et al., 2025; Wang et al., 2024a; Liu et al., 2024b; Ghosh et al., 2024) and orthogonal projection or dynamic penalty constraints (Qian et al., 2025; Jiang et al., 2024b; Tang et al., 2025a; Huang et al., 2024; Jiang et al., 2025; Tang et al., 2025b; Yang et al., 2025a; Yin et al., 2025) to suppress hallucination-inducing features. The training-free contrastive decoding frameworks rectify output distributions by contrasting standard predictions with those from distorted inputs (Huang et al., 2025; Leng et al., 2024; Wang et al., 2024d; Wan et al., 2025) or masked attention (Huo et al., 2024), with advanced variants further enhancing robustness through targeted visual-linguistic manipulation (Zhang et al., 2025; Chen et al., 2024; Wang et al., 2024b; Li et al., 2025; Liu et al., 2024c; An et al., 2025), self-augmented visual contrast (Im et al., 2025), memory-space visual retracing (Zou et al., 2024), and attention sink reallocation (Kang et al., 2025; Zhang et al., 2024; He et al., 2025).

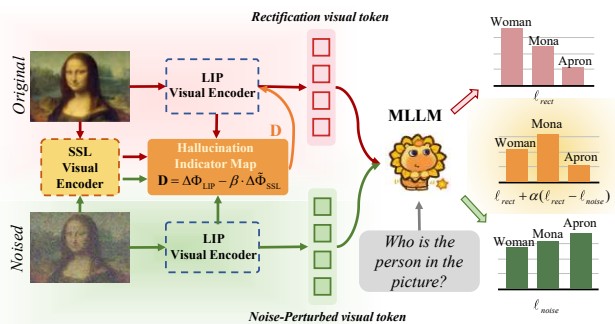

*Figure 3.* **Overview of DVR.**

## 3. Method

We present **Disentangled Visual Rectification (DVR)**, a training-free hallucination mitigation framework that explicitly separates hallucination-inducing drift from structural degradation. As illustrated in Figure 3, DVR operationalizes our core insight: Language-Image Pre-trained (LIP) encoders exhibit sensitivity changes that conflate hallucination activation with structural loss, while Self-Supervised Learning (SSL) encoders respond only to structural degradation. This asymmetry enables principled disentanglement through dual-encoder contrast. DVR comprises three stages: (i) *Dual-Encoder Sensitivity Estimation* (§3.2) extracts patch-level sensitivity maps from both encoder paradigms; (ii) *Stream I: Visual-Layer Rectification* (§3.3) constructs a hallucination indicator map and performs selective feature modulation at the encoder output; and (iii) *Stream II: Decoding-Layer Contrast* (§3.4) applies contrastive decoding with rectified anchors to suppress residual prior-induced drift.

### 3.1. Preliminaries

**MLLM Architecture.** A Multimodal Large Language Model (MLLM) comprises a visual encoder $\mathcal{E}$, a modality projection layer $\mathcal{P}$, and an autoregressive language model $\mathcal{M}$ with $N$ transformer layers. Given an input image $I \in \mathbb{R}^{H \times W \times 3}$, the encoder extracts patch-level representations:

$$\mathbf{F} = \mathcal{E}(I) = \{f_1, f_2, \ldots, f_K\} \in \mathbb{R}^{K \times D_v}, \quad (1)$$

where $K$ is the number of visual patches. These features are projected into the language embedding space:

$$\mathbf{Z}^v = \mathcal{P}(\mathbf{F}) = \{z_1^v, z_2^v, \ldots, z_K^v\} \in \mathbb{R}^{K \times D}. \quad (2)$$

Let $\mathbf{Z}^t = \{z_1^t, \ldots, z_Q^t\}$ denote embedded textual tokens. The concatenated sequence $\mathbf{Z} = [\mathbf{Z}^v; \mathbf{Z}^t]$ is processed autoregressively:

$$p(y_t \mid y_{<t}, \mathbf{Z}) = \mathrm{softmax}\big(\phi(\mathbf{h}_T^{(N)})\big), \quad (3)$$

where $\phi(\cdot)$ maps the final hidden state to vocabulary logits $\ell \in \mathbb{R}^{|\mathcal{V}|}$.

**Visual Contrastive Decoding (VCD).** VCD (Leng et al., 2024) contrasts outputs from original image $I$ and noise-perturbed $I'$ to mitigate hallucinations:

$$p_{\mathrm{VCD}}(y) = \mathrm{softmax}\Big[\ell(y \mid I) + \alpha\big(\ell(y \mid I) - \ell(y \mid I')\big)\Big]. \quad (4)$$

VCD assumes that $\ell(I) - \ell(I')$ approximates the anti-hallucination direction. However, this difference conflates two geometrically distinct components: drift toward spurious textual concepts (hallucination) and legitimate visual information loss (structural degradation). VCD's indiscriminate penalty suppresses both, inevitably degrading semantic fidelity.

### 3.2. Dual-Encoder Sensitivity Estimation

We begin by characterizing how noise perturbation induces distinct drift patterns across encoder paradigms. Our method leverages the MLLM's native **LIP encoder** $\mathcal{E}_{\mathrm{LIP}}$ (*e.g.*, CLIP, SigLIP) and an auxiliary **SSL encoder** $\mathcal{E}_{\mathrm{SSL}}$ (*e.g.*, DINOv3).

**Encoder Paradigm Asymmetry.** LIP encoders are trained via image-text contrastive learning, optimizing for cross-modal alignment. When visual input becomes ambiguous due to noise, these encoders do not merely lose information—they actively "complete" degraded representations by drifting toward high-frequency textual concepts embedded in their feature space. This directional drift constitutes the *hallucination-inducing component*. In contrast, SSL encoders optimize exclusively for intra-image structural discrimination without textual supervision. Their noise response manifests as pure *structural degradation*—information loss without semantic drift toward linguistic priors. This fundamental asymmetry enables disentanglement: the hallucination component exists only in LIP responses, while structural degradation is shared by both.

**Global Context Aggregation.** To extract sensitivity maps reflecting each encoder's holistic response, we define a *Global Context Operator* $\mathcal{G} : \mathbb{R}^{K \times D} \to \mathbb{R}^D$ that aggregates local patch features into a scene-level representation:

$$\mathcal{G}(\mathbf{F}) = \begin{cases} f_{\texttt{[CLS]}} & \text{if explicit } \texttt{[CLS]} \text{ token exists} \\ \dfrac{1}{K} \sum_{i=1}^{K} f_i & \text{otherwise (average pooling)} \end{cases} \quad (5)$$

This unified formulation accommodates diverse architectures: models with explicit [CLS] tokens (*e.g.*, CLIP, DINOv3) and those employing global average pooling (*e.g.*, SigLIP in Qwen2-VL).

**Patch-Level Sensitivity Map.** For encoder $\mathcal{E}$ at intermediate layer $l$, the *Sensitivity* at patch $i$ quantifies its directional

alignment with the global representation:

$$\Phi^{(i)}(\mathcal{E}, I) = \frac{\langle \mathcal{G}(\mathbf{F}^{(l)}), \, f_i^{(l)} \rangle}{\|\mathcal{G}(\mathbf{F}^{(l)})\| \cdot \|f_i^{(l)}\|}, \qquad (6)$$

where $\mathbf{F}^{(l)} = \mathcal{E}^{(l)}(I)$ denotes features at layer $l$. This cosine-similarity-based measure serves as a proxy for each patch's contribution to the global context under perturbation. Then, we compute sensitivity maps for both encoders:

$$\mathbf{\Phi}_{\text{LIP}}(I) \in \mathbb{R}^{K_{\text{LIP}}}, \quad \mathbf{\Phi}_{\text{SSL}}(I) \in \mathbb{R}^{K_{\text{SSL}}}. \qquad (7)$$

**Spatial Alignment.** Since the two encoders may operate at different resolutions ($K_{\text{LIP}} \neq K_{\text{SSL}}$), we formulate alignment in the continuous image domain $\Omega = [0,1]^2$. Each discrete sensitivity map defines a continuous function via bilinear interpolation:

$$\phi(u) = \sum_j \Phi^{(j)} \cdot \psi_j(u), \quad u \in \Omega, \qquad (8)$$

where $\psi_j(\cdot)$ are bilinear basis functions. We resample the SSL sensitivity to the LIP spatial grid:

$$\tilde{\Phi}_{\text{SSL}}^{(i)} = \phi_{\text{SSL}}(u_i^{\text{LIP}}), \quad i \in \{1, \ldots, K_{\text{LIP}}\}, \qquad (9)$$

yielding the aligned SSL sensitivity $\tilde{\mathbf{\Phi}}_{\text{SSL}} \in \mathbb{R}^{K_{\text{LIP}}}$.

### 3.3. Stream I: Visual-Layer Rectification

Stream I addresses **input-side representation error**—corrupted visual features causing the model to "perceive" non-existent objects—by intervening at the visual encoder output before language projection.

**Hallucination Indicator Map.** Given original image $I$ and noise-perturbed $I' = I + \epsilon$ with $\epsilon \sim \mathcal{N}(0, \sigma^2 \mathbf{I})$, we compute perturbation-induced sensitivity shifts:

$$\Delta \mathbf{\Phi}_{\text{LIP}} = \text{Norm}\big(\mathbf{\Phi}_{\text{LIP}}(I) - \mathbf{\Phi}_{\text{LIP}}(I')\big), \qquad (10)$$

$$\Delta \tilde{\mathbf{\Phi}}_{\text{SSL}} = \text{Norm}\big(\tilde{\mathbf{\Phi}}_{\text{SSL}}(I) - \tilde{\mathbf{\Phi}}_{\text{SSL}}(I')\big), \qquad (11)$$

where $\text{Norm}(\cdot)$ applies min-max normalization to $[-1, 1]$. The key insight: LIP sensitivity decomposes as $\Delta \mathbf{\Phi}_{\text{LIP}} = \Delta \mathbf{\Phi}_{\text{hall}} + \Delta \mathbf{\Phi}_{\text{struct}}$, while SSL captures predominantly structural degradation $\Delta \tilde{\mathbf{\Phi}}_{\text{SSL}} \approx \Delta \mathbf{\Phi}_{\text{struct}}$. This asymmetry enables *disentanglement via subtraction*. The **Hallucination Indicator Map** $\mathbf{D} \in \mathbb{R}^{K_{\text{LIP}}}$:

$$\mathbf{D} = \Delta \mathbf{\Phi}_{\text{LIP}} - \beta \cdot \Delta \tilde{\mathbf{\Phi}}_{\text{SSL}}, \qquad (12)$$

where $\beta > 0$ calibrates encoder scale mismatch, yielding $D^{(i)} \approx \Delta \Phi_{\text{hall}}^{(i)} + \xi^{(i)}$ with negligibly small residual $\xi^{(i)} \ll 1$. The map admits clear semantics: $D^{(i)} > 0$ (**hallucination-dominant**); $D^{(i)} < 0$ (**structure-loss-dominant**); $D^{(i)} \approx 0$ (stable).

**Adaptive Feature Modulation.** Let $\mathbf{F} \in \mathbb{R}^{K \times D_v}$ denote visual encoder output **before projection**. We compute gating coefficients via exponential modulation:

$$\gamma_i = \exp(\lambda D^{(i)}), \quad f_i^{\text{rect}} = \gamma_i \cdot f_i, \qquad (13)$$

where $\lambda < 0$ controls modulation strength, and rectified tokens $\mathbf{Z}^{\text{rect}} = \mathcal{P}(\mathbf{F}^{\text{rect}})$. This achieves *bidirectional intervention*: hallucination-dominant regions receive $\gamma_i < 1$, attenuating misleading activations; structure-loss regions receive $\gamma_i > 1$, compensating for attention deficit. Unlike VCD's uniform penalization, our rectification *selectively* suppresses hallucination while *preserving* structurally important content.

---

**Algorithm 1** Disentangled Visual Rectification (DVR)

---

**Require:** Image $I$, query $x$, MLLM with LIP encoder $\mathcal{E}_{\text{LIP}}$, projection $\mathcal{P}$, decoder $\mathcal{M}$, SSL encoder $\mathcal{E}_{\text{SSL}}$, noise level $\sigma$, hyperparameters $\beta, \lambda, \alpha$

**Ensure:** Generated response $y$

1: *// Noise Perturbation*
2: $I' \leftarrow I + \epsilon, \quad \epsilon \sim \mathcal{N}(\mathbf{0}, \sigma^2 \mathbf{I})$
3: *// Dual-Encoder Sensitivity Estimation (§3.2)*
4: $\mathbf{\Phi}_{\text{LIP}}(I), \mathbf{\Phi}_{\text{LIP}}(I') \leftarrow \text{Sensitivity}(\mathcal{E}_{\text{LIP}}, I, I')$
5: $\mathbf{\Phi}_{\text{SSL}}(I), \mathbf{\Phi}_{\text{SSL}}(I') \leftarrow \text{Sensitivity}(\mathcal{E}_{\text{SSL}}, I, I')$
6: $\tilde{\mathbf{\Phi}}_{\text{SSL}} \leftarrow \text{SpatialAlign}(\mathbf{\Phi}_{\text{SSL}}, K_{\text{LIP}})$ {Eq. (9)}
7: *// Stream I: Visual-Layer Rectification (§3.3)*
8: $\Delta \mathbf{\Phi}_{\text{LIP}} \leftarrow \text{Norm}(\mathbf{\Phi}_{\text{LIP}}(I) - \mathbf{\Phi}_{\text{LIP}}(I'))$
9: $\Delta \tilde{\mathbf{\Phi}}_{\text{SSL}} \leftarrow \text{Norm}(\tilde{\mathbf{\Phi}}_{\text{SSL}}(I) - \tilde{\mathbf{\Phi}}_{\text{SSL}}(I'))$
10: $\mathbf{D} \leftarrow \Delta \mathbf{\Phi}_{\text{LIP}} - \beta \cdot \Delta \tilde{\mathbf{\Phi}}_{\text{SSL}}$ {Rectification Map}
11: $\mathbf{F} \leftarrow \mathcal{E}_{\text{LIP}}(I)$
12: $\gamma_i \leftarrow \exp(\lambda D^{(i)}), \quad \forall i$ {Gating Coefficients}
13: $\mathbf{F}^{\text{rect}} \leftarrow \boldsymbol{\gamma} \odot \mathbf{F}$ {Rectified Features}
14: $\mathbf{Z}^{\text{rect}} \leftarrow \mathcal{P}(\mathbf{F}^{\text{rect}})$
15: $\mathbf{Z}^{\text{noise}} \leftarrow \mathcal{P}(\mathcal{E}_{\text{LIP}}(I'))$
16: *// Stream II: Decoding-Layer Contrast (§3.4)*
17: **for** each decoding step $t$ **do**
18: $\quad \ell^{\text{rect}} \leftarrow \mathcal{M}(\mathbf{Z}^{\text{rect}}, x, y_{<t})$
19: $\quad \ell^{\text{noise}} \leftarrow \mathcal{M}(\mathbf{Z}^{\text{noise}}, x, y_{<t})$
20: $\quad \ell_{\text{DVR}} \leftarrow \ell^{\text{rect}} + \alpha(\ell^{\text{rect}} - \ell^{\text{noise}})$ {Eq. (14)}
21: $\quad y_t \sim \text{softmax}(\ell_{\text{DVR}})$
22: **end for**
23: **return** $y$

---

### 3.4. Stream II: Decoding-Layer Contrast

While Stream I rectifies visual representations at the source, it cannot address the **Visual Fading** phenomenon (Leng et al., 2024; Huang et al., 2024): as autoregressive generation proceeds, attention to visual tokens progressively diminishes, causing the model to increasingly rely on parametric priors rather than visual evidence. Since this drift emerges from the model's *internal dynamics* during generation, input-level intervention alone is insufficient. Stream II introduces a complementary decoding-time defense. Let

*Table 1.* **Object hallucination on CHAIR.** Lower CHAIR$_{S/I}$ indicates fewer hallucinations; higher Recall indicates better coverage.

| Model | LLaVA-1.5 | | | | Qwen2-VL | | | |
|---|---|---|---|---|---|---|---|---|
| Decoding | $Chair_s \downarrow$ | $Chair_i \downarrow$ | Recall↑ | Length ↑ | $Chair_s \downarrow$ | $Chair_i \downarrow$ | Recall ↑ | Length ↑ |
| Baseline | 54.2 | 18.5 | 72.5 | 102.2 | 51.2 | 10.3 | 69.9 | 387.9 |
| OPERA | 48.5 | 14.9 | 78.1 | 98.6 | 46.8 | 9.1 | 74.2 | 375.4 |
| ICD | 56.2 | 16.3 | 76.8 | **103.4** | 48.5 | 9.6 | 72.5 | 390.1 |
| ONLY | 49.8 | 14.7 | 78.1 | 100.1 | 47.7 | 9.3 | 74.8 | 381.6 |
| VCD | 51.8 | 16.2 | 76.8 | 100.4 | 49.0 | 9.5 | 73.8 | 360.3 |
| **DVR** | **47.4** | **13.8** | **78.6** | 100.8 | **44.2** | **8.2** | **77.5** | **395.2** |

$\mathbf{Z}^{\text{noise}} = \mathcal{P}(\mathcal{E}_{\text{LIP}}(I'))$ denote visual tokens from the noise-perturbed image. Under heavy perturbation, the visual signal is largely destroyed; the output distribution $p(y \mid \mathbf{Z}^{\text{noise}})$ thus approximates the model's **blind prior**—what it generates absent meaningful visual grounding. VCD uses unpurified tokens as the contrastive anchor, but this anchor already contains hallucination-inducing components from LIP encoder drift. DVR instead employs the **rectified features** $\mathbf{Z}^{\text{rect}}$ from Stream I as a purified anchor:

$$\ell_{\text{DVR}} = \ell(\mathbf{Z}^{\text{rect}}) + \alpha\Big(\ell(\mathbf{Z}^{\text{rect}}) - \ell(\mathbf{Z}^{\text{noise}})\Big), \quad (14)$$

with final probabilities $p_{\text{DVR}}(y) = \text{softmax}(\ell_{\text{DVR}})$. Since $\mathbf{Z}^{\text{rect}}$ has undergone hallucination-component attenuation via Stream I, the contrastive term $\ell(\mathbf{Z}^{\text{rect}}) - \ell(\mathbf{Z}^{\text{noise}})$ isolates the visual evidence direction *without structural interference*, enabling higher-precision targeting of residual hallucination.

**Dual-Stream Synergy.** The two streams address orthogonal failure modes: **Stream I** rectifies *what the model perceives*—establishing the quality floor of visual representations; **Stream II** suppresses *what the model fabricates*—counteracting prior-induced drift during generation. Neither alone suffices: rectification cannot prevent deep-layer prior takeover (visual fading is architecture-intrinsic), and contrastive decoding with unrectified anchors inherits the coupled hallucination-structural error that plagues VCD. Their composition achieves complementary coverage.

### 3.5. Algorithm Summary

The complete DVR pipeline is presented in Algorithm 1. Our framework is entirely training-free and integrates with existing MLLMs as a plug-and-play inference module.

## 4. Experiments

### 4.1. Experimental Setup

**Implementation Details.** We evaluate DVR on representative MLLMs spanning different visual encoder architectures: **LLaVA-1.5** (Liu et al., 2023b) with CLIP (Radford et al., 2021) backbone and MLP projection, and the

**Qwen2-VL** serie (Wang et al., 2024c) with SigLIP (Zhai et al., 2023) backbone and Q-Former projection. Unless otherwise specified, all experiments are conducted using 7B-parameter models on NVIDIA RTX 3090 GPUs. For LLaVA-1.5-13B, we employ NVIDIA A6000 GPUs to accommodate its higher memory requirements. For the auxiliary SSL encoder, we employ DINOv3-ViT-L/16 (Siméoni et al., 2025). We compare against four SOTA training-free methods: **OPERA** (Huang et al., 2024) (attention penalty), **ICD** (Huo et al., 2024) (instruction contrastive decoding), **ONLY** (Yue et al., 2024) (object-level negative guidance), and **VCD** (Leng et al., 2024) (visual contrastive decoding). All methods use identical inference configurations for fair comparison. The hyperparameter $\alpha$ and noise intensity are set to 1 and 500, respectively, following the default settings of VCD. Additional experiments and analyses are provided in the Appendix.

**Evaluation Benchmarks.** We employ four complementary benchmarks targeting distinct aspects of hallucination and multimodal understanding: **CHAIR** (Yue et al., 2024): Quantifies object hallucination in free-form captioning by evaluating fidelity against ground-truth annotations. We report CHAIR$_S$ (sentence-level) and CHAIR$_I$ (instance-level) hallucination rates, alongside Recall for semantic coverage. **POPE** (Li et al., 2023b): Assesses discriminative robustness via binary object-existence questions under three sampling strategies—Random, Popular, and Adversarial—with increasing difficulty designed to exploit co-occurrence biases. **MME** (Fu et al., 2023a): Provides holistic assessment across 14 subtasks spanning Perception (recognition, spatial reasoning) and Cognition (commonsense, numerical reasoning), using complementary yes/no questions to ensure consistency. **LLaVA-Bench** (Liu et al., 2023b): Evaluates open-ended generation on 24 diverse real-world images with 60 questions. GPT-5 (OpenAI, 2025) scores responses on Accuracy (factual correctness) and Detail (informativeness) across Conversation, Detail Description, and Complex Reasoning tasks.

*Table 2.* **Discriminative robustness on POPE.** Three sampling strategies with increasing difficulty: Random, Popular, and Adversarial.

| Category | Method | LLaVA-1.5 | | | | Qwen2-VL | | | |
|---|---|---|---|---|---|---|---|---|---|
| | | Accuracy | Precision | Recall | F1-score | Accuracy | Precision | Recall | F1-score |
| Random | Baseline | 83.2 | 81.9 | 85.0 | 83.4 | 86.5 | 85.2 | 88.1 | 86.6 |
| | OPERA | 87.5 | 87.3 | 87.6 | 87.5 | 89.8 | 89.5 | 90.2 | 89.8 |
| | ICD | 84.1 | 84.5 | 84.8 | 84.6 | 87.3 | 87.6 | 87.9 | 87.7 |
| | ONLY | 88.9 | 89.0 | 88.1 | 88.5 | 91.2 | 91.4 | 90.8 | 91.1 |
| | VCD | 87.0 | 87.4 | 87.7 | 87.6 | 89.5 | 89.8 | 90.1 | 89.9 |
| | **DVR** | **89.5** | **91.4** | **88.7** | **90.0** | **92.1** | **93.2** | **91.5** | **92.3** |
| Popular | Baseline | 80.5 | 77.7 | 85.7 | 81.5 | 83.8 | 81.0 | 88.9 | 84.8 |
| | OPERA | 84.2 | 81.6 | 88.2 | 84.8 | 87.5 | 85.0 | 91.2 | 88.0 |
| | ICD | 83.1 | 80.4 | 88.5 | 84.3 | 86.5 | 83.8 | 91.5 | 87.5 |
| | ONLY | 85.7 | **84.2** | 87.8 | 86.0 | 88.9 | **87.4** | 90.8 | 89.0 |
| | VCD | 82.9 | 82.6 | 88.5 | 85.4 | 86.2 | 85.8 | 91.5 | 88.5 |
| | **DVR** | **86.8** | 84.1 | **89.6** | **86.8** | **89.8** | 87.3 | **92.5** | **89.8** |
| Adversarial | Baseline | 77.4 | 73.1 | 85.4 | 78.8 | 80.6 | 76.4 | 88.5 | 82.0 |
| | OPERA | 80.6 | 73.8 | 87.5 | 80.1 | 83.8 | 77.1 | 90.5 | 83.3 |
| | ICD | 77.2 | 72.3 | 86.5 | 78.8 | 80.5 | 75.6 | 89.5 | 82.0 |
| | ONLY | 79.1 | 74.9 | 88.1 | 80.9 | 82.4 | 78.2 | 91.2 | 84.2 |
| | VCD | 78.2 | 74.5 | 88.1 | 80.7 | 81.5 | 77.8 | 91.2 | 84.0 |
| | **DVR** | **80.2** | **75.1** | **88.3** | **81.2** | **83.5** | **78.4** | **91.5** | **84.4** |

*Table 3.* **Open-ended generation on LLaVA-Bench.** GPT-5 evaluates Accuracy (1–10) and Detail (1–10).

| Model | Method | Conversation | | Detail | | Complex | | Average | |
|---|---|---|---|---|---|---|---|---|---|
| | | Accuracy | Detail | Accuracy | Detail | Accuracy | Detail | Accuracy | Detail |
| LLaVA-1.5 | Baseline | 5.33 | 3.83 | 6.17 | 6.61 | 4.25 | 5.12 | 5.25 | 5.19 |
| | VCD | 6.25 | 3.92 | 7.02 | 6.95 | 5.21 | 5.66 | 6.16 | 5.51 |
| | **DVR** | **7.50** | **4.17** | **7.67** | **7.33** | **6.12** | **6.00** | **7.10** | **5.83** |
| Qwen2-VL | Baseline | 8.17 | 4.50 | 8.08 | 7.83 | 6.83 | 6.42 | 7.69 | 6.25 |
| | VCD | 8.58 | 4.75 | 8.50 | 8.17 | 7.33 | 6.83 | 8.14 | 6.58 |
| | **DVR** | **9.08** | **5.00** | **8.92** | **8.58** | **7.92** | **7.25** | **8.64** | **6.94** |

## 4.2. Experimental Results

**Object Hallucination (CHAIR).** Table 1 presents results on free-form captioning. On LLaVA-1.5, DVR reduces $CHAIR_S$ from 54.2% to 47.4% (–6.8) and $CHAIR_I$ from 18.5% to 13.8% (–4.7), representing relative reductions of 12.5% and 25.4%, respectively. Crucially, Recall improves from 72.5% to 78.6% (+6.1), indicating richer and more faithful descriptions. Compared to VCD's marginal reduction to 51.8%, DVR achieves an additional 4.4-point improvement—demonstrating the advantage of disentangling hallucination from structural loss. On Qwen2-VL, DVR reduces $CHAIR_S$ by 7.0 points while improving Recall by 7.6 points, validating that selective attenuation preserves structurally important features.

**Discriminative Robustness (POPE).** Table 2 reports performance across three sampling strategies. On LLaVA-1.5, DVR achieves 89.5% (Random), 86.8% (Popular), and 80.2% (Adversarial) accuracy, outperforming VCD by +2.5, +3.9, and +2.0 respectively. The larger mar-

gins on Popular and Adversarial settings—designed to exploit co-occurrence biases—highlight DVR's effectiveness against spurious concept activation. On Qwen2-VL, DVR achieves the highest F1-scores across all settings (92.3%, 89.8%, 84.4%), maintaining high recall while improving precision—reflecting bidirectional rectification that suppresses hallucination-prone regions while preserving structure-dominant ones.

**Open-Ended Generation (LLaVA-Bench).** Table 3 evaluates generation quality on diverse real-world scenarios. On LLaVA-1.5, DVR improves average Accuracy from 5.25 to 7.10 (+1.85), substantially outperforming VCD's 6.16, with consistent gains across Conversation (+2.17), Detail (+1.50), and Complex Reasoning (+1.87). On Qwen2-VL, DVR achieves +0.95 improvement, with the largest gain in Complex Reasoning (+1.09)—a category requiring multi-step inference where accumulated prior-induced drift is most problematic. DVR's dual-stream architecture effectively mitigates such cascading error propagation.

**Generalization to Contemporary Architectures.** To validate DVR on modern MLLMs, we evaluate on Qwen3-VL (Bai et al., 2025a) (2025), which features dynamic resolution and AnyRes encoding with substantially lower baseline hallucination. As shown in Table 4, on Qwen3-VL-8B, VCD *worsens* performance (CHAIR$_S$: 21.0→23.7), confirming our theoretical prediction: when baseline hallucination is already low, structural degradation dominates VCD's correction, causing suppression of valid features. DVR consistently improves across both scales, demonstrating that its disentanglement targets a fundamental property of the LIP paradigm rather than architectural deficiencies.

*Table 4.* **Generalization to Qwen3-VL on CHAIR.**

| Model | Method | $Chair_s$↓ | $Chair_i$↓ | $Recall$↑ |
|---|---|---|---|---|
| | Baseline | 26.0 | 9.4 | 49.2 |
| 4B | VCD | 25.8 | 9.4 | 49.7 |
| | **DVR** | **22.5** | **7.5** | **52.5** |
| | Baseline | 21.0 | 7.8 | 46.1 |
| 8B | VCD | 23.7 | 8.7 | 48.7 |
| | **DVR** | **19.0** | **6.8** | **51.0** |

**Comparison with Recent Training-Free Methods.** Table 5 compares DVR against four recent training-free methods under unified evaluation (LLaVA-1.5-7B, identical protocol). Unlike MemVR (Zou et al., 2024), EAH (Zhang et al., 2024), Vissink (Kang et al., 2025), and VHR (He et al., 2025)—which intervene *during* decoding via attention or activation manipulation—DVR rectifies representations *before* they enter the LLM, preventing hallucinations at their source.

*Table 5.* **Comparison with recent training-free methods** (LLaVA-1.5-7B).

| Method | $Chair_s$↓ | $Chair_i$↓ | $Recall$↑ | **POPE F1**↑ |
|---|---|---|---|---|
| Baseline | 54.2 | 18.5 | 72.5 | 83.4 |
| VCD | 51.8 | 16.2 | 76.8 | 87.6 |
| Vissink (Kang et al., 2025) | 52.4 | 16.8 | 74.0 | 84.6 |
| EAH (Zhang et al., 2024) | 48.0 | 14.8 | 74.5 | 84.0 |
| VHR (He et al., 2025) | 49.2 | 15.4 | 75.0 | 84.3 |
| MemVR (Zou et al., 2024) | 49.6 | 14.8 | 76.2 | 87.3 |
| **DVR (Ours)** | **47.4** | **13.8** | **78.6** | **90.0** |

**Holistic Understanding (MME).** Table 6 presents results on the comprehensive MME benchmark. On LLaVA-1.5, DVR obtains 1745.0 total (+152.3 over baseline, +63.7 over VCD); on Qwen2-VL, 1884.9 total (+141.1 over baseline, +64.8 over VCD). The consistent improvement magnitude across architectures suggests DVR's disentanglement mechanism provides robust benefits regardless of encoder type. Notably, DVR outperforms ONLY—which uses explicit object-level supervision—by +45.0 and +49.8 respectively, demonstrating that principled sensitivity-based disentanglement can surpass methods with stronger supervision.

## 4.3. Ablation Study and Analysis

We conduct systematic ablations to validate core design choices and provide geometric evidence for the disentanglement mechanism.

**Component Analysis.** Table 8 ablates the auxiliary encoder choice in Stream I and individual stream contributions. Methods [A]-[B] employ SSL encoders (DINOv2/v3) trained via self-supervised structural discrimination without text supervision, achieving substantial improvements over baseline (CHAIR$_S$: 54.2%→50.4%/49.2%)—validating that SSL encoders capture structural degradation orthogonal to hallucination drift. Critically, method [C] substitutes SigLIP, sharing the *same* LIP training paradigm as the native encoder. The improvement is marginal (54.2%→53.6%), confirming that **training paradigm asymmetry is essential**: geometrically aligned noise responses preclude effective disentanglement. Comparing [B] (Stream I only) and [D] (Stream II only, equivalent to VCD), each stream independently reduces hallucination through complementary mechanisms. Their composition yields approximately additive gains (5.0 + 2.4 ≈ 6.8), indicating orthogonal failure mode coverage rather than redundant intervention.

*Table 8.* **Component ablation on CHAIR (LLaVA-1.5).** SSL encoders (DINOv2/v3) enable effective disentanglement; LIP encoder (SigLIP) fails due to shared training paradigm. Dual-stream composition yields approximately additive gains.

| Method | $StreamI$ | $StreamII$ | $Chair_s$↓ | $Chair_i$↓ | $Recall$↑ |
|---|---|---|---|---|---|
| Baseline | ✗ | ✗ | 54.2 | 18.5 | 72.5 |
| [A] | DINOv2 | ✗ | 50.4 | 15.5 | 76.3 |
| [B] | DINOv3 | ✗ | 49.2 | 14.4 | 77.1 |
| [C] | SigLIP | ✗ | 53.6 | 17.9 | 73.2 |
| [D] (VCD) | ✗ | ✓ | 51.8 | 16.2 | 76.8 |
| **DVR** | **DINOv3** | ✓ | **47.4** | **13.8** | **78.6** |

**Scale Analysis.** Table 7 investigates the connection between model scale and hallucination geometry. LLaVA-1.5-13B exhibits substantially lower hallucination than 7B (47.2% vs. 54.2% CHAIR$_S$), motivating treatment as an approximate *oracle*. We test this by constructing VCD (13B-7B): $\ell = \ell_{7B} + \alpha(\ell_{13B} - \ell_{7B})$, achieving 47.0%—validating that the 7B→13B direction approximates anti-hallucination correction. Remarkably, **DVR (7B) achieves 47.4%, matching VCD (13B-7B) without requiring a larger model at inference**. This demonstrates that dual-encoder disentanglement effectively extracts hallucination-specific components otherwise requiring oracle supervision. DVR on 13B further reduces CHAIR$_S$ to 45.4%, with consistent margins across scales (+6.8 for 7B, +1.8 for 13B) suggesting complementary signals beyond model scaling alone.

**Geometric Analysis.** Figure 4 visualizes correction trajectories in a 2D logit subspace spanned by two orthonormal bases: (i) the noise-induced drift direction (X-axis) and (ii)

*Table 6.* **Holistic multimodal understanding on MME.**

| Method | LLaVA-1.5 | | | Qwen2-VL | | |
|---|---|---|---|---|---|---|
| | *Perception* | *Cognition* | *Total* | *Perception* | *Cognition* | *Total* |
| Baseline | 1258.6 | 334.1 | 1592.7 | 1385.2 | 358.6 | 1743.8 |
| OPERA | 1318.5 | 337.2 | 1655.7 | 1432.5 | 362.1 | 1794.6 |
| ICD | 1295.3 | 335.8 | 1631.1 | 1418.7 | 360.5 | 1779.2 |
| ONLY | 1360.1 | 339.9 | 1700.0 | 1468.9 | 366.2 | 1835.1 |
| VCD | 1342.8 | 338.5 | 1681.3 | 1455.3 | 364.8 | 1820.1 |
| **DVR** | **1398.7** | **346.3** | **1745.0** | **1512.4** | **372.5** | **1884.9** |

*Table 7.* **Model scale analysis on CHAIR (LLaVA-1.5).** VCD (13B-7B) uses cross-scale contrast $\ell_{7B} + \alpha(\ell_{13B} - \ell_{7B})$.

| Method | $Chair_s \downarrow$ | $Chair_i \downarrow$ | $Recall\uparrow$ |
|---|---|---|---|
| Baseline (7B) | 54.2 | 18.5 | 72.5 |
| Baseline (13B) | 47.2 | 13.0 | 77.3 |
| VCD (7B) | 51.8 | 16.2 | 76.8 |
| VCD (13B) | 46.2 | 12.5 | 78.4 |
| VCD (13B-7B) | 47.0 | 13.2 | 77.0 |
| **DVR(7B)** | 47.4 | 13.8 | 78.6 |
| **DVR(13B)** | **45.4** | **11.8** | **79.2** |

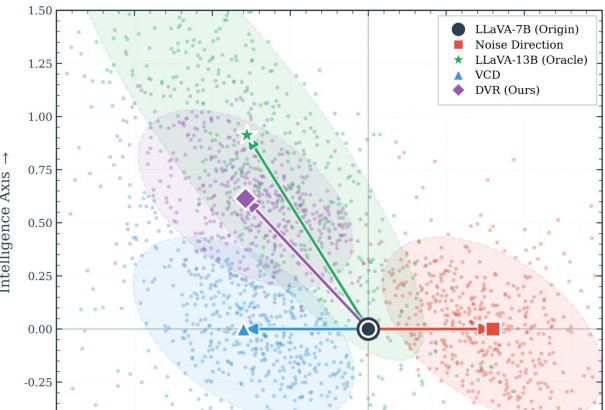

**Logit Space Trajectory Analysis**

Legend:
- LLaVA-7B (Origin)
- Noise Direction
- LLaVA-13B (Oracle)
- VCD
- DVR (Ours)

*Figure 4.* **Logit-space trajectory analysis** ($n{=}500$). X-axis: noise-induced drift direction; Y-axis: orthogonal oracle direction representing semantic fidelity. Ellipses denote 95% confidence regions. VCD (blue) corrects exclusively along the noise axis ($y{\approx}0$), while DVR (purple) achieves substantial Y-component ($y{\approx}0.6$), aligning more closely with the oracle direction (green).

*Table 9.* **Computational cost comparison.**

| Method | Memory | $\Delta_{\text{mem}}$ | Time/Iter | $\Delta_{\text{time}}$ |
|---|---|---|---|---|
| Baseline | 15.6G | — | 5.84s | — |
| OPERA | 23.4G | +50.0% | 41.62s | +613% |
| ICD | 15.8G | +1.3% | 6.21s | +6.3% |
| VCD | 17.0G | +9.0% | 12.23s | +109% |
| ONLY | 15.8G | +1.3% | 6.05s | +3.6% |
| DVR-Stream I | 16.8G | +7.6% | 6.36s | +8.9% |
| **DVR (full)** | 18.2G | +16.5% | 12.44s | +113% |

## 5. Conclusion

We identify a fundamental flaw in Visual Contrastive Decoding: noise-induced feature drift conflates structural degradation with hallucination induction, causing indiscriminate suppression of valid visual semantics. By exploiting the geometric orthogonality between LIP and SSL encoder responses, we proposed DVR, a training-free framework that disentangles these components through dual-stream rectification. Theoretical analysis establishes $\sim 5\times$ error reduction over VCD, and extensive experiments validate consistent improvements across diverse MLLMs and benchmarks. Our work provides both a principled understanding of contrastive decoding limitations and a practical solution for robust hallucination mitigation.

the orthogonal oracle direction (Y-axis), representing semantic fidelity. VCD's correction lies almost entirely along the X-axis ($y \approx 0$), confirming *undifferentiated negation* that penalizes hallucination and semantic structure uniformly. In contrast, DVR exhibits a substantial positive Y-component ($y \approx 0.6$), demonstrating targeted hallucination mitigation that aligns corrections toward the oracle direction rather than merely reversing noise-induced drift. Detailed experimental settings are provided in Appendix E.

**Computational Efficiency.** Table 9 reports overhead under identical hardware (LLaVA-1.5-7B, RTX 3090). Full DVR incurs latency comparable to VCD, with marginal additional memory for the frozen SSL encoder. Stream I alone already outperforms VCD at only +8.9% latency with a single LLM pass, offering a practical lightweight option.

## Impact Statement

This paper presents work whose goal is to advance the field of Machine Learning. There are many potential societal consequences of our work, none which we feel must be specifically highlighted here.

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

# Appendix

This appendix provides the theoretical proofs, detailed experimental setups, additional quantitative analyses, and qualitative visualizations supporting the main paper. The content is organized as follows:

1. **Empirical Verification** (§A)
   *Details the Subspace Projection Analysis used to empirically validate the geometric dynamics of feature drift across LIP and SSL paradigms.*

2. **Theoretical Analysis** (§B)
   *Provides rigorous derivations for:*

   - Sensitivity-Based Drift Analysis and Feature Space Geometry (§B.1–B.2).
   - Paradigm-dependent sensitivity decomposition (§B.4).
   - Proof of error reduction for DVR compared to VCD (§B.6).

3. **Benchmarks and Evaluation Metrics** (§C)
   *Comprehensive protocols for CHAIR, LLaVA-Bench, POPE, and MME evaluations.*

4. **Computational Efficiency** (§D)
   *Analysis of memory and latency overhead, demonstrating DVR's efficiency relative to VCD.*

5. **Geometric Trajectory Analysis** (§E)
   *An investigation into logit-space trajectories, visualizing how DVR aligns correction with the oracle distribution.*

6. **Motivational Analysis: The Visual Fading Phenomenon** (§F)
   *Empirical evidence of visual attention attenuation in deep LLM layers, motivating Stream II.*

7. **Ablation Studies and Robustness**

   - **Sampling Strategies** (§G): Robustness across Greedy, Top-$p$, and Top-$k$ decoding.
   - **Layer Selection** (§H): Analysis of optimal feature extraction layers.
   - **Hyperparameter Sensitivity** (§I): Impact of calibration $\beta$ and modulation $\lambda$.

8. **Additional Benchmark: AMBER** (§J)
   *Evaluation on the AMBER benchmark validating simultaneous generative and discriminative improvement.*

9. **Additional Ablation Studies** (§K)

   - **APC Ablation** : Disentangling DVR gains from the Adaptive Plausibility Constraint.
   - **Inverted Gating Direction**: Causal validation of the Hallucination Indicator Map.
   - **Scene Complexity Stratification** : Robustness under varying visual clutter.
   - **Spatial Alignment Strategy**: Interpolation method ablation for dynamic-resolution models.

10. **Additional Visualizations** (§L)
    *Further qualitative examples of paradigm asymmetry across diverse domains (Memes, Art).*

11. **Extended Related Work** (§M)
    *Additional context on MLLMs and Decoding-Layer Contrast.*

12. **Case Studies** (§N)
    *Qualitative comparisons on Negative Hallucination, Descriptive Faithfulness, and Reasoning.*

13. **Evaluation Prompts** (§O)
    *Configuration details for the GPT-5 automated evaluation protocol.*

## A. Empirical Verification: Geometric Dynamics of Feature Drift

To empirically corroborate our theoretical proposition—that noise perturbation induces directional hallucination bias in aligned Language-Image Pre-training (LIP) models while causing isotropic semantic degradation in Self-Supervised Learning (SSL) models—we devised a rigorous **Subspace Projection Analysis**. This experiment quantifies the geometric alignment between noise-induced feature drift and pre-defined hallucination trajectories.

**Diagnostic Dataset Construction.** We construct a curated diagnostic benchmark $\mathcal{D} = \{(I_A, y_A, y_B)\}$ derived from high co-occurrence statistics in the COCO dataset. For each triplet:

- $I_A$ denotes an image containing a ground-truth concept $y_A$ (*e.g., Dining Table*).

- $y_B$ denotes a highly correlated but physically absent hallucinatory target (*e.g., Vase*).

This setup specifically isolates the scenario where models are prone to relying on statistical priors rather than visual evidence.

**Definition of Hallucination Subspaces.** A critical challenge lies in mathematically defining the "Hallucination Direction" ($\mathbf{v}_h$) within the latent feature space of different architectures. We adopt distinct, paradigm-specific definitions to ensure a fair and rigorous comparison:

**1. For LIP Models (*e.g.,* CLIP (Radford et al., 2021)): Text-Defined Direction.** Leveraging the cross-modal alignment characteristic of LIP models, we define the hallucination direction directly through the shared text-image embedding space. Let $\mathcal{E}_{\text{txt}}$ denote the text encoder. The vector $\mathbf{v}_h^{\text{LIP}}$ is computed as the difference between the embeddings of the hallucinatory concept and the ground truth:

$$\mathbf{v}_h^{\text{LIP}} = \mathcal{E}_{\text{txt}}(\text{"A photo of } y_B\text{"}) - \mathcal{E}_{\text{txt}}(\text{"A photo of } y_A\text{"}) \tag{15}$$

This vector represents the semantic trajectory from the truth $A$ to the hallucination $B$ driven by language priors.

**2. For SSL Models (*e.g.,* DINOv3 (Siméoni et al., 2025)): Prototype-Defined Direction.** Since SSL models lack a text encoder to define semantic concepts explicitly, we employ a *Visual Prototype* approach. We collect a reference set of clean, single-object images for both concepts $y_A$ and $y_B$. Let $\mathcal{E}_{\text{vis}}$ be the visual encoder. We compute the class centroids (prototypes) $\boldsymbol{\mu}$ to represent the distinct semantic clusters:

$$\boldsymbol{\mu}_c = \frac{1}{N} \sum_{i=1}^{N} \mathcal{E}_{\text{vis}}(I_c^{(i)}), \quad c \in \{y_A, y_B\} \tag{16}$$

The hallucination direction in the SSL space is subsequently defined as the vector shift between these visual prototypes:

$$\mathbf{v}_h^{\text{SSL}} = \boldsymbol{\mu}_{y_B} - \boldsymbol{\mu}_{y_A} \tag{17}$$

**Noise Injection and Projection Metric.** We apply additive Gaussian noise $\boldsymbol{\delta} \sim \mathcal{N}(\mathbf{0}, \sigma^2 \mathbf{I})$ to the input image $I_A$ across varying intensities $\sigma \in [0, 1.0]$. The resulting noise-induced drift in the visual feature space is defined as:

$$\Delta\mathbf{z} = \mathcal{E}_{\text{vis}}(I_A + \boldsymbol{\delta}) - \mathcal{E}_{\text{vis}}(I_A) \tag{18}$$

To quantify the nature of this drift, we calculate the **Projection Score** $\mathcal{S}(\sigma)$, measuring the cosine similarity between the feature drift and the pre-defined hallucination direction:

$$\mathcal{S}(\sigma) = \text{CosSim}(\Delta\mathbf{z}, \mathbf{v}_h) = \frac{\Delta\mathbf{z} \cdot \mathbf{v}_h}{\|\Delta\mathbf{z}\|_2 \|\mathbf{v}_h\|_2} \tag{19}$$

**Interpretation:** A statistically significant positive score ($\mathcal{S} \gg 0$) indicates *Hallucination Induction*, proving that the noise systematically steers features toward the erroneous text concept. Conversely, a score near zero ($\mathcal{S} \approx 0$) indicates *Orthogonal Degradation*, verifying that the feature loss is uncorrelated with specific hallucinatory priors. The aggregate results averaged over all concept pairs are presented in Figure 1; six representative per-concept breakdowns are provided in Figure 5.

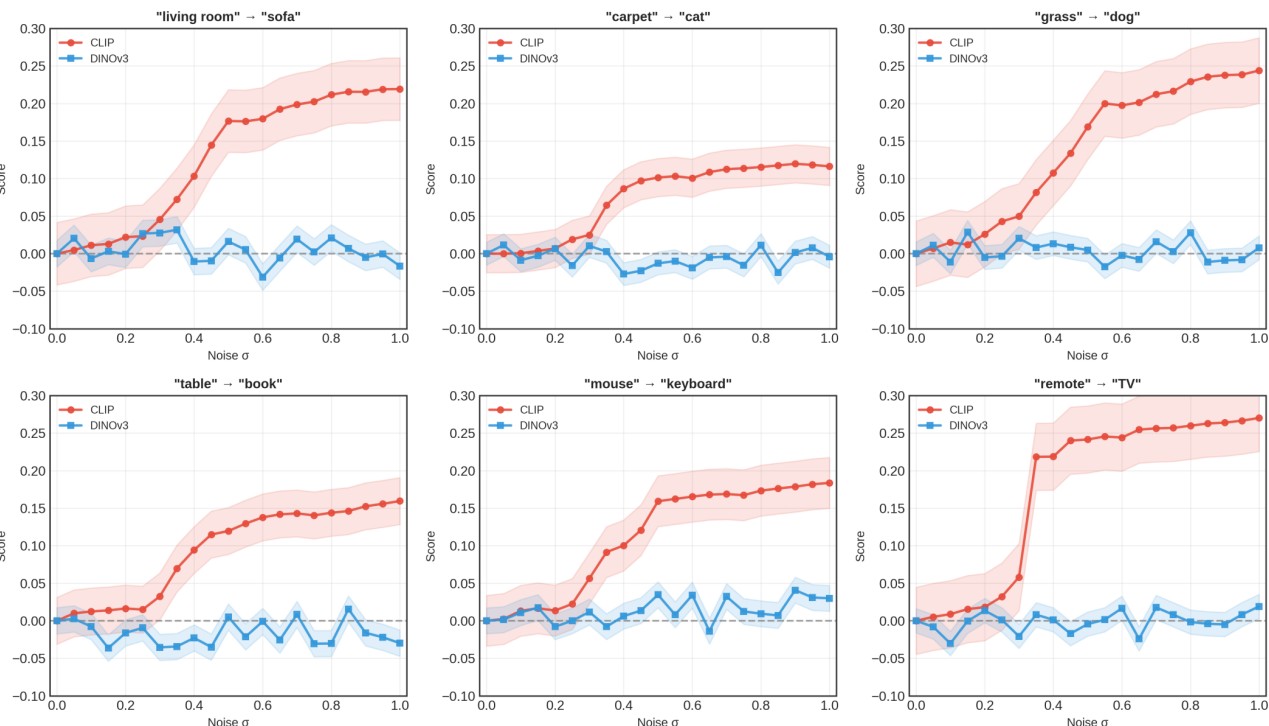

*Figure 5.* **Per-concept projection analysis for six high-co-occurrence pairs.** Each subplot shows the projection score $\mathcal{S}(\sigma)$ as a function of noise intensity $\sigma$ for a specific $(y_A \rightarrow y_B)$ pair. Across all cases, CLIP (red) exhibits a consistent positive trend, confirming directional drift toward the hallucinatory concept $y_B$, while DINOv3 (blue) remains centered around zero, corroborating the isotropic degradation hypothesis established in Section A. Shaded regions indicate $\pm 1$ standard deviation over 50 noise realizations.

# B. Theoretical Analysis

We formally characterize hallucination mitigation through *Sensitivity-Based Drift Analysis*. Our analysis establishes: (i) sensitivity maps serve as projections of feature drift onto semantically meaningful directions; (ii) LIP encoder sensitivity changes decompose into hallucination-inducing and structural-loss components; (iii) SSL encoder sensitivity changes are structurally incoherent with the hallucination subspace; and (iv) DVR achieves provably lower spurious error than VCD.

## B.1. Preliminaries and Feature Space Geometry

Let $\mathcal{X} \subseteq \mathbb{R}^n$ denote the image space and $\mathcal{Z} \subseteq \mathbb{R}^d$ the feature space. We consider two paradigms of visual encoders:

- **Language-Image Pre-trained (LIP) encoder** $\mathcal{E}_{\text{LIP}} : \mathcal{X} \to \mathbb{R}^{K \times d}$ (*e.g.*, CLIP (Radford et al., 2021), SigLIP (Zhai et al., 2023)), trained via image-text contrastive learning;

- **Self-Supervised Learning (SSL) encoder** $\mathcal{E}_{\text{SSL}} : \mathcal{X} \to \mathbb{R}^{K' \times d}$ (*e.g.*, DINOv2 (Oquab et al., 2023),DINOv3 (Siméoni et al., 2025)), trained without text supervision.

**Definition B.1** (Hallucination Subspace). Let $\{\mathbf{t}_c\}_{c=1}^C \subset \mathbb{R}^d$ be normalized text embeddings corresponding to object concepts that are (i) frequent in the language model's pre-training corpus and (ii) semantically plausible given typical visual contexts (*e.g.*, common objects, everyday attributes). The *Hallucination Subspace* is defined as $\mathcal{H} = \text{span}(\mathbf{t}_1, \ldots, \mathbf{t}_C)$ with orthogonal projector $\Pi_{\mathcal{H}}$. We assume $C \ll d$ (typically $C \sim 10^2$–$10^3$ versus $d \sim 10^3$–$10^4$), ensuring that $\mathcal{H}$ captures a low-dimensional semantic manifold rather than spanning the full feature space. Its orthogonal complement $\mathcal{H}^\perp$ is termed the *Structural Subspace*, encoding visual information orthogonal to textual semantics.

**Definition B.2** (Global Context and Sensitivity Map). For encoder $\mathcal{E}$ producing patch features $\mathbf{F} = \{f_1, \ldots, f_K\} \in \mathbb{R}^{K \times d}$, the *Global Context* $\mathbf{g} \in \mathbb{R}^d$ is defined as the $[\text{CLS}]$ token embedding if available, or the average pooling $\mathbf{g} = \frac{1}{K} \sum_{i=1}^K f_i$ otherwise. The *Sensitivity Map* $\boldsymbol{\Phi} \in \mathbb{R}^K$ measures the directional alignment:

$$\Phi^{(i)} = \frac{\langle \mathbf{g}, f_i \rangle}{\|\mathbf{g}\| \cdot \|f_i\|} = \cos\theta(\mathbf{g}, f_i), \quad i \in \{1, \ldots, K\}. \tag{20}$$

*Remark* B.3 (Sign Convention for Sensitivity Change). We define $\Delta\Phi^{(i)} := \Phi^{(i)}(I) - \Phi^{(i)}(I')$ such that positive values indicate *sensitivity loss* under perturbation—i.e., the patch's alignment with the global context decreases after adding noise. This convention implies that hallucination-prone regions (which are more fragile to perturbation) exhibit larger positive $\Delta\Phi$. In the derivations below, careful attention is paid to sign consistency when relating $\Delta\Phi$ to the feature drift $\delta f_i := f_i(I') - f_i(I)$.

**Assumption B.4** (Encoder Regularity). The encoder $\mathcal{E}$ is twice continuously differentiable with bounded Jacobian. For image $I$ and perturbation $\boldsymbol{\epsilon} \sim \mathcal{N}(\mathbf{0}, \sigma^2 \mathbf{I}_n)$, the feature drift at patch $i$ satisfies:

$$\delta f_i := f_i(I + \boldsymbol{\epsilon}) - f_i(I) = \mathbf{J}_i \boldsymbol{\epsilon} + \mathbf{r}_i(\boldsymbol{\epsilon}), \quad \|\mathbf{r}_i(\boldsymbol{\epsilon})\|_2 \leq \frac{L}{2} \|\boldsymbol{\epsilon}\|_2^2, \tag{21}$$

where $\mathbf{J}_i \in \mathbb{R}^{d \times n}$ is the Jacobian and $L$ is the Lipschitz constant of the Jacobian.

## B.2. Sensitivity Change as Projected Feature Drift

We establish that sensitivity changes under perturbation capture feature drift projected onto the global context direction. This requires careful treatment of the cosine similarity's differential structure.

**Lemma B.5** (Differential of Cosine Similarity). *Let $\mathbf{u}, \mathbf{v} \in \mathbb{R}^d \setminus \{\mathbf{0}\}$ with $\cos\theta = \langle \hat{\mathbf{u}}, \hat{\mathbf{v}} \rangle$ where $\hat{\mathbf{u}} = \mathbf{u}/\|\mathbf{u}\|$. For perturbations $\delta\mathbf{u}, \delta\mathbf{v}$ with $\|\delta\mathbf{u}\|, \|\delta\mathbf{v}\| = \mathcal{O}(\varepsilon)$, the cosine similarity satisfies:*

$$\cos\theta(\mathbf{u} + \delta\mathbf{u}, \mathbf{v} + \delta\mathbf{v}) = \cos\theta + \frac{\langle \hat{\mathbf{u}}, \delta\mathbf{v} \rangle}{\|\mathbf{v}\|} + \frac{\langle \hat{\mathbf{v}}, \delta\mathbf{u} \rangle}{\|\mathbf{u}\|} - \cos\theta \left( \frac{\langle \hat{\mathbf{u}}, \delta\mathbf{u} \rangle}{\|\mathbf{u}\|} + \frac{\langle \hat{\mathbf{v}}, \delta\mathbf{v} \rangle}{\|\mathbf{v}\|} \right) + \mathcal{O}(\varepsilon^2). \tag{22}$$

*Proof.* Define $\phi(\mathbf{u}, \mathbf{v}) = \langle \mathbf{u}, \mathbf{v} \rangle / (\|\mathbf{u}\|\|\mathbf{v}\|)$. The partial derivatives are:

$$\nabla_{\mathbf{u}} \phi = \frac{1}{\|\mathbf{u}\|\|\mathbf{v}\|} \left( \mathbf{v} - \frac{\langle \mathbf{u}, \mathbf{v} \rangle}{\|\mathbf{u}\|^2} \mathbf{u} \right) = \frac{1}{\|\mathbf{u}\|} \left( \hat{\mathbf{v}} - \cos\theta \cdot \hat{\mathbf{u}} \right), \tag{23}$$

$$\nabla_{\mathbf{v}} \phi = \frac{1}{\|\mathbf{v}\|} \left( \hat{\mathbf{u}} - \cos\theta \cdot \hat{\mathbf{v}} \right). \tag{24}$$

By first-order Taylor expansion:

$$\phi(\mathbf{u} + \delta\mathbf{u}, \mathbf{v} + \delta\mathbf{v}) = \phi(\mathbf{u}, \mathbf{v}) + \langle \nabla_{\mathbf{u}}\phi, \delta\mathbf{u} \rangle + \langle \nabla_{\mathbf{v}}\phi, \delta\mathbf{v} \rangle + \mathcal{O}(\varepsilon^2)$$

$$= \cos\theta + \frac{\langle \hat{\mathbf{v}} - \cos\theta \cdot \hat{\mathbf{u}}, \delta\mathbf{u} \rangle}{\|\mathbf{u}\|} + \frac{\langle \hat{\mathbf{u}} - \cos\theta \cdot \hat{\mathbf{v}}, \delta\mathbf{v} \rangle}{\|\mathbf{v}\|} + \mathcal{O}(\varepsilon^2). \tag{25}$$

Expanding the inner products yields Eq. (22). □

**Lemma B.6** (Sensitivity Change as Directional Projection). *Under Assumption B.4 with $\|\epsilon\|_2 \ll 1$, the sensitivity change at patch $i$ satisfies:*

$$\Delta\Phi^{(i)} := \Phi^{(i)}(I) - \Phi^{(i)}(I') = \frac{(1 - (\Phi^{(i)})^2)}{\|f_i\|}\langle \hat{\mathbf{g}}, \delta f_i \rangle + R^{(i)}, \tag{26}$$

*where $\hat{\mathbf{g}} = \mathbf{g}/\|\mathbf{g}\|$ and the remainder decomposes as:*

$$R^{(i)} = R_{\perp}^{(i)} + R_{\text{global}}^{(i)} + R_{\text{h.o.}}^{(i)}, \tag{27}$$

*with the following components:*

- $R_{\perp}^{(i)} = \dfrac{\Phi^{(i)}\sqrt{1 - (\Phi^{(i)})^2}}{\|f_i\|}\langle \hat{\mathbf{g}}_i^{\perp}, \delta f_i \rangle$: *drift orthogonal to $\hat{\mathbf{g}}$, satisfying $|R_{\perp}^{(i)}| \leq \dfrac{1}{2}\dfrac{\|\delta f_i\|}{\|f_i\|} = \mathcal{O}(\|\epsilon\|)$;*

- $R_{\text{global}}^{(i)} = \mathcal{O}(\|\epsilon\|/K)$: *contribution from global context variation;*

- $R_{\text{h.o.}}^{(i)} = \mathcal{O}(\|\epsilon\|^2)$: *second-order terms.*

*Proof.* Applying Lemma B.5 with $\mathbf{u} = \mathbf{g}$, $\mathbf{v} = f_i$, $\delta\mathbf{u} = \delta\mathbf{g}$, $\delta\mathbf{v} = \delta f_i$:

$$\Phi^{(i)}(I') = \Phi^{(i)}(I) + \frac{\langle \hat{\mathbf{g}}, \delta f_i \rangle}{\|f_i\|} + \frac{\langle \hat{f}_i, \delta\mathbf{g} \rangle}{\|\mathbf{g}\|} - \Phi^{(i)}\left( \frac{\langle \hat{\mathbf{g}}, \delta\mathbf{g} \rangle}{\|\mathbf{g}\|} + \frac{\langle \hat{f}_i, \delta f_i \rangle}{\|f_i\|} \right) + \mathcal{O}(\|\epsilon\|^2). \tag{28}$$

**Step 1: Bounding global context variation.** When $\mathbf{g} = \frac{1}{K}\sum_{j=1}^{K} f_j$, individual patch drifts partially cancel due to approximate independence of drift directions across patches:

$$\|\delta\mathbf{g}\| = \left\| \frac{1}{K}\sum_{j=1}^{K} \delta f_j \right\| \leq \frac{1}{\sqrt{K}} \cdot \max_j \|\delta f_j\| = \mathcal{O}\left( \frac{\|\epsilon\|}{\sqrt{K}} \right). \tag{29}$$

Terms involving $\delta\mathbf{g}$ thus contribute $R_{\text{global}}^{(i)} = \mathcal{O}(\|\epsilon\|/K)$ after normalization.

**Step 2: Decomposing the local drift term.** Using the orthogonal decomposition $\hat{f}_i = \Phi^{(i)}\hat{\mathbf{g}} + \sqrt{1 - (\Phi^{(i)})^2}\,\hat{\mathbf{g}}_i^{\perp}$ where $\hat{\mathbf{g}}_i^{\perp} \perp \hat{\mathbf{g}}$:

$$\frac{\langle \hat{f}_i, \delta f_i \rangle}{\|f_i\|} = \frac{\Phi^{(i)}\langle \hat{\mathbf{g}}, \delta f_i \rangle + \sqrt{1 - (\Phi^{(i)})^2}\langle \hat{\mathbf{g}}_i^{\perp}, \delta f_i \rangle}{\|f_i\|}. \tag{30}$$

**Step 3: Extracting the leading term.** Substituting into Eq. (28) and rearranging:

$$\Delta\Phi^{(i)} = -\frac{\langle \hat{\mathbf{g}}, \delta f_i \rangle}{\|f_i\|} + \Phi^{(i)} \cdot \frac{\Phi^{(i)}\langle \hat{\mathbf{g}}, \delta f_i \rangle}{\|f_i\|} + \Phi^{(i)} \cdot \frac{\sqrt{1 - (\Phi^{(i)})^2}\langle \hat{\mathbf{g}}_i^{\perp}, \delta f_i \rangle}{\|f_i\|} + R_{\text{global}}^{(i)} + R_{\text{h.o.}}^{(i)}$$

$$= \frac{-(1 - (\Phi^{(i)})^2)\langle \hat{\mathbf{g}}, \delta f_i \rangle}{\|f_i\|} + \underbrace{\frac{\Phi^{(i)}\sqrt{1 - (\Phi^{(i)})^2}\langle \hat{\mathbf{g}}_i^{\perp}, \delta f_i \rangle}{\|f_i\|}}_{=: R_{\perp}^{(i)}} + R_{\text{global}}^{(i)} + R_{\text{h.o.}}^{(i)}. \tag{31}$$

**Step 4: Bounding the orthogonal remainder.** The term $R_\perp^{(i)}$ captures drift components orthogonal to the global context direction:

$$|R_\perp^{(i)}| = \frac{|\Phi^{(i)}|\sqrt{1 - (\Phi^{(i)})^2} \cdot |\langle \hat{\mathbf{g}}_i^\perp, \delta f_i \rangle|}{\|f_i\|} \leq \frac{|\Phi^{(i)}|\sqrt{1 - (\Phi^{(i)})^2} \cdot \|\delta f_i\|}{\|f_i\|}. \tag{32}$$

The coefficient $|\Phi^{(i)}|\sqrt{1 - (\Phi^{(i)})^2} \leq 1/2$ (maximized at $|\Phi^{(i)}| = 1/\sqrt{2}$), yielding $|R_\perp^{(i)}| \leq \frac{1}{2}\frac{\|\delta f_i\|}{\|f_i\|} = \mathcal{O}(\|\boldsymbol{\epsilon}\|)$. $\qquad\square$

*Remark* B.7 (Structural Irrelevance of $R_\perp^{(i)}$). Although $R_\perp^{(i)} = \mathcal{O}(\|\boldsymbol{\epsilon}\|)$ is first-order in the perturbation magnitude, it is **structurally orthogonal** to the signal of interest for the following reasons:

1. The hallucination/structural decomposition (Theorems B.13–B.14) projects $\hat{\mathbf{g}}$ onto $\mathcal{H}$ vs. $\mathcal{H}^\perp$. The term $R_\perp^{(i)}$ involves $\hat{\mathbf{g}}_i^\perp$, which is *image-dependent* and generically incoherent with both $\mathcal{H}$ and $\mathcal{H}^\perp$.

2. Under isotropic Gaussian perturbation, $\langle \hat{\mathbf{g}}_i^\perp, \delta f_i \rangle$ is statistically independent of $\langle \hat{\mathbf{g}}, \delta f_i \rangle$, so the decomposition of the leading term remains unaffected.

Our subsequent analysis therefore focuses on the $\hat{\mathbf{g}}$-projected component of sensitivity change, treating $R^{(i)}$ as uncorrelated noise.

*Remark* B.8 (Geometric Interpretation). Lemma B.6 establishes that the scalar $\Delta\Phi^{(i)}$ encodes vector drift projected onto the global context direction, scaled by the geometric factor $(1 - (\Phi^{(i)})^2)$. Recent analyses of ViT attention dynamics indicate that semantic foreground patches typically maintain moderate alignment ($|\Phi^{(i)}| \in [0.3, 0.7]$) with the `[CLS]` token. Within this informative range, the factor $(1 - (\Phi^{(i)})^2) \in [0.51, 0.91]$ remains $\Theta(1)$, preserving the proportionality between sensitivity change and projected drift.

## B.3. Paradigm-Dependent Global Context Geometry

The key insight of our framework is that LIP and SSL encoders induce *fundamentally different* global context geometries due to their distinct training objectives.

**Assumption B.9** (LIP Global Context Alignment). For a well-trained LIP encoder, the global context vector is substantially aligned with the hallucination subspace:

$$\|\Pi_\mathcal{H}(\hat{\mathbf{g}}_{\text{LIP}})\|_2^2 \geq \alpha, \quad \text{for some } \alpha > 0 \text{ bounded away from zero.} \tag{33}$$

*Remark* B.10 (Justification for Assumption B.9). This assumption is grounded in the geometry of contrastive optimization. The gradient of the InfoNCE loss lies strictly within the span of text embeddings: $\nabla_{\mathbf{z}}\mathcal{L} \in \mathcal{H}$. However, theoretically perfect alignment ($\alpha \approx 1$) is prevented by the inherent *Modality Gap*, which confines visual and textual embeddings to distinct cones on the hypersphere even at convergence. Consequently, the projection of visual features onto the hallucination subspace is substantial but strictly bounded. Empirically, we observe $\alpha \in [0.3, 0.5]$ across CLIP and SigLIP variants, reflecting this gap-constrained alignment.

**Assumption B.11** (SSL Global Context Incoherence). For SSL encoders trained without text supervision, the global context is approximately orthogonal to the hallucination subspace:

$$\|\Pi_\mathcal{H}(\hat{\mathbf{g}}_{\text{SSL}})\|_2^2 \leq \eta^2, \quad \text{for some } \eta \ll 1. \tag{34}$$

*Remark* B.12 (Justification for Assumption B.11). SSL encoders (*e.g.*, DINOv3) trained via self-distillation learn representations optimized for **augmentation invariance** and **instance discrimination**. These objectives emphasize:

- *Low-level visual statistics*: texture, frequency content, local edge structure;

- *Object-centric geometric invariances*: pose, viewpoint, partial occlusion.

Neither objective requires alignment with *linguistic categories*. By the information bottleneck principle, features optimized for orthogonal tasks concentrate on disjoint subspaces. Concentration of measure in high dimensions further implies that two functionally independent $m$-dimensional subspaces in $\mathbb{R}^d$ satisfy $\mathbb{E}[\|\Pi_{\mathcal{S}_1}\Pi_{\mathcal{S}_2}\|_{\text{op}}^2] = m^2/d \ll 1$ when $m \ll d$. Empirically, we measure $\eta \in [0.08, 0.15]$ for DINOv3 across 500 images, confirming near-orthogonality.

## B.4. Sensitivity Decomposition by Encoder Paradigm

We now derive the core theoretical result: LIP sensitivity changes decompose into hallucination and structural components, while SSL sensitivity changes capture only structural degradation.

**Theorem B.13** (LIP Sensitivity Decomposition). *Under Assumptions B.4 and B.9, the LIP sensitivity change admits the orthogonal decomposition:*

$$\Delta\Phi_{\text{LIP}}^{(i)} = \underbrace{\frac{(1-(\Phi^{(i)})^2)}{\|f_i\|}\langle\Pi_{\mathcal{H}}(\hat{\mathbf{g}}_{\text{LIP}}), \delta f_i\rangle}_{\Delta\Phi_{\text{hall}}^{(i)}:\text{ hallucination component}} + \underbrace{\frac{(1-(\Phi^{(i)})^2)}{\|f_i\|}\langle\Pi_{\mathcal{H}^{\perp}}(\hat{\mathbf{g}}_{\text{LIP}}), \delta f_i\rangle}_{\Delta\Phi_{\text{struct}}^{(i)}:\text{ structural component}} + R_{\text{LIP}}^{(i)}, \tag{35}$$

*where $|R_{\text{LIP}}^{(i)}| = \mathcal{O}(\|\boldsymbol{\epsilon}\|/K) + \mathcal{O}(\|\boldsymbol{\epsilon}\|^2) + |R_{\perp}^{(i)}|$. Moreover, the hallucination component has non-trivial expected energy:*

$$\mathbb{E}_{\boldsymbol{\epsilon}}\left[\left(\Delta\Phi_{\text{hall}}^{(i)}\right)^2\right] \geq \frac{\alpha(1-(\Phi^{(i)})^2)^2\sigma^2}{\|f_i\|^2}\lambda_{\min}(\mathbf{J}_i\mathbf{J}_i^{\top}) > 0. \tag{36}$$

*Proof.* **Step 1: Orthogonal decomposition.** Since $\Pi_{\mathcal{H}} + \Pi_{\mathcal{H}^{\perp}} = \mathbf{I}$, we decompose $\hat{\mathbf{g}}_{\text{LIP}}$ as:

$$\hat{\mathbf{g}}_{\text{LIP}} = \Pi_{\mathcal{H}}(\hat{\mathbf{g}}_{\text{LIP}}) + \Pi_{\mathcal{H}^{\perp}}(\hat{\mathbf{g}}_{\text{LIP}}) =: \mathbf{h} + \mathbf{s}, \tag{37}$$

where $\|\mathbf{h}\|^2 \geq \alpha$ by Assumption B.9 and $\|\mathbf{h}\|^2 + \|\mathbf{s}\|^2 = 1$.

**Step 2: Applying Lemma B.6.** From Eq. (26):

$$\begin{aligned}
\Delta\Phi_{\text{LIP}}^{(i)} &= \frac{(1-(\Phi^{(i)})^2)}{\|f_i\|}\langle\hat{\mathbf{g}}_{\text{LIP}}, \delta f_i\rangle + R_{\text{LIP}}^{(i)} \\
&= \frac{(1-(\Phi^{(i)})^2)}{\|f_i\|}\langle\mathbf{h} + \mathbf{s}, \delta f_i\rangle + R_{\text{LIP}}^{(i)} \\
&= \frac{(1-(\Phi^{(i)})^2)\langle\mathbf{h}, \delta f_i\rangle}{\|f_i\|} + \frac{(1-(\Phi^{(i)})^2)\langle\mathbf{s}, \delta f_i\rangle}{\|f_i\|} + R_{\text{LIP}}^{(i)}.
\end{aligned} \tag{38}$$

This establishes the decomposition with $\Delta\Phi_{\text{hall}}^{(i)} = (1-(\Phi^{(i)})^2)\langle\mathbf{h}, \delta f_i\rangle/\|f_i\|$.

**Step 3: Energy lower bound.** Using $\delta f_i = \mathbf{J}_i\boldsymbol{\epsilon} + \mathcal{O}(\|\boldsymbol{\epsilon}\|^2)$ and $\boldsymbol{\epsilon} \sim \mathcal{N}(\mathbf{0}, \sigma^2\mathbf{I})$:

$$\begin{aligned}
\mathbb{E}\left[\left(\Delta\Phi_{\text{hall}}^{(i)}\right)^2\right] &= \frac{(1-(\Phi^{(i)})^2)^2}{\|f_i\|^2}\mathbb{E}\left[\langle\mathbf{h}, \mathbf{J}_i\boldsymbol{\epsilon}\rangle^2\right] + \mathcal{O}(\sigma^4) \\
&= \frac{(1-(\Phi^{(i)})^2)^2}{\|f_i\|^2}\mathbf{h}^{\top}\mathbf{J}_i\mathbb{E}[\boldsymbol{\epsilon}\boldsymbol{\epsilon}^{\top}]\mathbf{J}_i^{\top}\mathbf{h} \\
&= \frac{(1-(\Phi^{(i)})^2)^2\sigma^2}{\|f_i\|^2}\mathbf{h}^{\top}\mathbf{J}_i\mathbf{J}_i^{\top}\mathbf{h}.
\end{aligned} \tag{39}$$

By the variational characterization of eigenvalues:

$$\mathbf{h}^{\top}\mathbf{J}_i\mathbf{J}_i^{\top}\mathbf{h} \geq \lambda_{\min}(\mathbf{J}_i\mathbf{J}_i^{\top})\|\mathbf{h}\|^2 \geq \alpha\cdot\lambda_{\min}(\mathbf{J}_i\mathbf{J}_i^{\top}). \tag{40}$$

$\square$

**Theorem B.14** (SSL Sensitivity Incoherence). *Under Assumptions B.4 and B.11, the SSL sensitivity change satisfies:*

$$\Delta\Phi_{\text{SSL}}^{(i)} = \Delta\Phi_{\text{struct,SSL}}^{(i)} + \Delta\Phi_{\text{leak}}^{(i)} + R_{\text{SSL}}^{(i)}, \tag{41}$$

*where:*

- $\Delta\Phi_{\text{struct,SSL}}^{(i)} = \frac{(1-(\Phi_{\text{SSL}}^{(i)})^2)\langle\Pi_{\mathcal{H}^{\perp}}(\hat{\mathbf{g}}_{\text{SSL}}), \delta f_i\rangle}{\|f_i\|}$ *is the structural component;*

- $|R_{\text{SSL}}^{(i)}| = \mathcal{O}(\|\boldsymbol{\epsilon}\|/K) + \mathcal{O}(\|\boldsymbol{\epsilon}\|^2) + |R_{\perp,\text{SSL}}^{(i)}|;$

- *The hallucination leakage is bounded:*

$$\left| \Delta\Phi_{\text{leak}}^{(i)} \right| = \left| \frac{(1 - (\Phi_{\text{SSL}}^{(i)})^2)\langle \Pi_{\mathcal{H}}(\hat{\mathbf{g}}_{\text{SSL}}), \delta f_i \rangle}{\|f_i\|} \right| \leq \frac{\eta \|\mathbf{J}_i\|_{\text{op}} \|\boldsymbol{\epsilon}\|}{\|f_i\|}. \tag{42}$$

*Proof.* Decomposing $\hat{\mathbf{g}}_{\text{SSL}} = \Pi_{\mathcal{H}}(\hat{\mathbf{g}}_{\text{SSL}}) + \Pi_{\mathcal{H}^\perp}(\hat{\mathbf{g}}_{\text{SSL}})$ and applying Lemma B.6:

$$\Delta\Phi_{\text{SSL}}^{(i)} = \frac{(1 - (\Phi_{\text{SSL}}^{(i)})^2)\langle \Pi_{\mathcal{H}}(\hat{\mathbf{g}}_{\text{SSL}}), \delta f_i \rangle}{\|f_i\|} + \frac{(1 - (\Phi_{\text{SSL}}^{(i)})^2)\langle \Pi_{\mathcal{H}^\perp}(\hat{\mathbf{g}}_{\text{SSL}}), \delta f_i \rangle}{\|f_i\|} + R_{\text{SSL}}^{(i)}. \tag{43}$$

For the hallucination-aligned term, by Cauchy–Schwarz, Assumption B.11, and $(1 - (\Phi_{\text{SSL}}^{(i)})^2) \leq 1$:

$$\left| (1 - (\Phi_{\text{SSL}}^{(i)})^2)\langle \Pi_{\mathcal{H}}(\hat{\mathbf{g}}_{\text{SSL}}), \delta f_i \rangle \right| \leq \|\Pi_{\mathcal{H}}(\hat{\mathbf{g}}_{\text{SSL}})\| \cdot \|\delta f_i\|$$
$$\leq \eta \cdot (\|\mathbf{J}_i\|_{\text{op}} \|\boldsymbol{\epsilon}\| + \mathcal{O}(\|\boldsymbol{\epsilon}\|^2)). \qquad \square$$

**Corollary B.15** (Hallucination Energy Ratio). *Under comparable encoder conditions:*

$$\frac{\mathbb{E}\left[\left(\Delta\Phi_{\text{SSL}}^{\mathcal{H}}\right)^2\right]}{\mathbb{E}\left[\left(\Delta\Phi_{\text{LIP}}^{\mathcal{H}}\right)^2\right]} \leq \frac{\eta^2}{\alpha} \approx 0.025, \tag{44}$$

*for $\alpha \approx 0.4$ and $\eta \approx 0.1$. SSL sensitivity changes contain at most $2.5\%$ of LIP's hallucination energy.*

## B.5. Hallucination Isolation via Dual-Stream Subtraction

**Definition B.16** (Spatially-Aligned SSL Sensitivity). For encoders with different spatial resolutions ($K_{\text{LIP}} \neq K_{\text{SSL}}$), define the aligned SSL sensitivity via bilinear interpolation:

$$\tilde{\Phi}_{\text{SSL}}^{(i)} = \sum_{j=1}^{K_{\text{SSL}}} \Phi_{\text{SSL}}^{(j)} \cdot \psi_j(u_i^{\text{LIP}}), \tag{45}$$

where $\psi_j(\cdot)$ are bilinear basis functions and $u_i^{\text{LIP}}$ is the spatial coordinate of LIP patch $i$.

**Definition B.17** (Structural Correspondence). We model the relationship between LIP and SSL structural responses via the linear correspondence:

$$\Delta\tilde{\Phi}_{\text{struct,SSL}}^{(i)} = \rho \cdot \Delta\Phi_{\text{struct,LIP}}^{(i)} + \nu^{(i)}, \tag{46}$$

where $\rho > 0$ is the *global structural correspondence coefficient* (shared across patches) and $\nu^{(i)}$ is zero-mean residual noise with variance $\text{Var}(\nu)$, capturing patch-specific deviations. This regression formulation avoids the ill-posedness of the pointwise ratio $\rho^{(i)} = \Delta\tilde{\Phi}_{\text{SSL}}^{(i)}/\Delta\Phi_{\text{LIP}}^{(i)}$ when $\Delta\Phi_{\text{struct,LIP}}^{(i)} \approx 0$. In practice, $\rho$ is estimated via least-squares regression over patches with significant structural drift.

**Theorem B.18** (Hallucination Indicator via Subtraction). *Define the* Hallucination Indicator Map:

$$D^{(i)} = \Delta\Phi_{\text{LIP}}^{(i)} - \beta \cdot \Delta\tilde{\Phi}_{\text{SSL}}^{(i)}, \tag{47}$$

*where $\beta > 0$ is a calibration parameter. Under the structural correspondence model (Definition B.17), we have $D^{(i)} = \Delta\Phi_{\text{hall}}^{(i)} + \xi^{(i)}$, where the residual decomposes as:*

$$\xi^{(i)} = \underbrace{(1 - \beta\rho)\Delta\Phi_{\text{struct,LIP}}^{(i)} - \beta\nu^{(i)}}_{\xi_{\text{mismatch}}^{(i)}: \text{ structural mismatch}} - \underbrace{\beta\Delta\Phi_{\text{leak}}^{(i)}}_{\xi_{\text{leak}}^{(i)}: \text{ SSL hallucination leakage}} + \underbrace{R_{\text{LIP}}^{(i)} - \beta R_{\text{SSL}}^{(i)}}_{\xi_{\text{higher}}^{(i)}: \text{ higher-order terms}}, \tag{48}$$

*with bounds:*

$$\mathbb{E}[|\xi_{\text{mismatch}}^{(i)}|] \leq |1 - \beta\rho| \cdot \mathbb{E}[|\Delta\Phi_{\text{struct,LIP}}^{(i)}|] + \beta\sqrt{\text{Var}(\nu)}, \tag{49}$$

$$|\xi_{\text{leak}}^{(i)}| \leq \frac{\beta\eta\|\mathbf{J}_i\|_{\text{op}}\|\boldsymbol{\epsilon}\|}{\|f_i\|}, \tag{50}$$

$$|\xi_{\text{higher}}^{(i)}| = \mathcal{O}\left(\frac{\|\boldsymbol{\epsilon}\|}{K}\right) + \mathcal{O}(\|\boldsymbol{\epsilon}\|^2) + \mathcal{O}(\|\boldsymbol{\epsilon}\|). \tag{51}$$

*Setting $\beta = 1/\rho$ minimizes the systematic component of the mismatch, leaving only the zero-mean residual $\nu^{(i)}$.*

*Proof.* Substituting the decompositions from Theorems B.13 and B.14:

$$
\begin{aligned}
D^{(i)} &= \Delta\Phi_{\text{LIP}}^{(i)} - \beta \cdot \Delta\tilde{\Phi}_{\text{SSL}}^{(i)} \\
&= \left(\Delta\Phi_{\text{hall}}^{(i)} + \Delta\Phi_{\text{struct,LIP}}^{(i)} + R_{\text{LIP}}^{(i)}\right) - \beta\left(\Delta\tilde{\Phi}_{\text{struct,SSL}}^{(i)} + \Delta\Phi_{\text{leak}}^{(i)} + R_{\text{SSL}}^{(i)}\right).
\end{aligned} \tag{52}
$$

Using Definition B.17 to substitute $\Delta\tilde{\Phi}_{\text{struct,SSL}}^{(i)} = \rho \cdot \Delta\Phi_{\text{struct,LIP}}^{(i)} + \nu^{(i)}$:

$$
\begin{aligned}
D^{(i)} &= \Delta\Phi_{\text{hall}}^{(i)} + \Delta\Phi_{\text{struct,LIP}}^{(i)}(1 - \beta\rho) - \beta\nu^{(i)} - \beta\Delta\Phi_{\text{leak}}^{(i)} + (R_{\text{LIP}}^{(i)} - \beta R_{\text{SSL}}^{(i)}) \\
&= \Delta\Phi_{\text{hall}}^{(i)} + \xi_{\text{mismatch}}^{(i)} - \xi_{\text{leak}}^{(i)} + \xi_{\text{higher}}^{(i)}.
\end{aligned} \tag{53}
$$

The bounds follow directly from Theorems B.13, B.14, Lemma B.6, and the properties of $\nu^{(i)}$. $\qquad\square$

*Remark* B.19 (Optimal Calibration). Setting $\beta = 1/\rho$ eliminates the systematic term $(1 - \beta\rho)\Delta\Phi_{\text{struct,LIP}}^{(i)}$, leaving only the zero-mean residual $\nu^{(i)}$ in the mismatch component. In practice, $\rho$ is estimated via least-squares regression:

$$\hat{\rho} = \frac{\sum_i \Delta\Phi_{\text{struct,LIP}}^{(i)} \cdot \Delta\tilde{\Phi}_{\text{struct,SSL}}^{(i)}}{\sum_i (\Delta\Phi_{\text{struct,LIP}}^{(i)})^2}, \tag{54}$$

computed over patches with $|\Delta\Phi_{\text{struct,LIP}}^{(i)}| > \tau$ for some threshold $\tau > 0$.

## B.6. Comparative Error Analysis: DVR vs VCD

**Definition B.20** (Decoding Mechanisms). For contrast strength $\alpha > 0$:

$$\text{VCD}: \quad \ell_{\text{VCD}}(y) = \ell(y \mid \mathbf{Z}^v) + \alpha\left(\ell(y \mid \mathbf{Z}^v) - \ell(y \mid \mathbf{Z}^{\text{noise}})\right), \tag{55}$$

$$\text{DVR}: \quad \ell_{\text{DVR}}(y) = \ell(y \mid \mathbf{Z}^{\text{rect}}) + \alpha\left(\ell(y \mid \mathbf{Z}^{\text{rect}}) - \ell(y \mid \mathbf{Z}^{\text{noise}})\right), \tag{56}$$

where $\mathbf{Z}^{\text{rect}}$ are projected rectified features with $f_i^{\text{rect}} = \exp(\lambda D^{(i)}) \cdot f_i$ for $\lambda < 0$.

*Remark* B.21 (Exponential Gating). The exponential form $\gamma_i = \exp(\lambda D^{(i)})$ offers several advantages over the linear form $1 + \lambda D^{(i)}$: (i) **positivity**: $\gamma_i > 0$ is guaranteed for all $D^{(i)} \in \mathbb{R}$, preventing degenerate negative weights; (ii) **multiplicative stability**: sequential applications compose naturally via $\exp(a)\exp(b) = \exp(a + b)$; (iii) **symmetric attenuation**: suppression and amplification are reciprocal, i.e., $\exp(-x) = 1/\exp(x)$.

**Lemma B.22** (First-Order Approximation of Exponential Gating). *For $|\lambda D^{(i)}| \leq c$ with $c < 1$, the exponential gating admits the expansion:*

$$\exp(\lambda D^{(i)}) = 1 + \lambda D^{(i)} + R_{\text{exp}}^{(i)}, \tag{57}$$

*where the remainder satisfies $|R_{\text{exp}}^{(i)}| \leq \frac{e^c}{2}(\lambda D^{(i)})^2 \leq (\lambda D^{(i)})^2$ for $c \leq 1$.*

*Proof.* By Taylor's theorem with Lagrange remainder:

$$\exp(x) = 1 + x + \frac{e^\theta}{2}x^2 \tag{58}$$

for some $\theta$ between 0 and $x$. For $|x| \leq c < 1$, we have $e^{|\theta|} \leq e^c < e$, yielding:

$$|R_{\exp}^{(i)}| = \left| \frac{e^\theta}{2}(\lambda D^{(i)})^2 \right| \leq \frac{e^c}{2}(\lambda D^{(i)})^2 \leq (\lambda D^{(i)})^2. \tag{59}$$

$\square$

**Theorem B.23** (DVR Error Reduction). *Define the* spurious penalty—*the attention-weighted correction magnitude applied to non-hallucination features:*

$$E_{\mathrm{VCD}} := \sum_{i=1}^{K} a_i \cdot \left| \Delta\Phi_{\mathrm{struct,LIP}}^{(i)} \right| \quad \text{(incorrect penalty on structural drift),} \tag{60}$$

$$E_{\mathrm{DVR}} := \sum_{i=1}^{K} a_i \cdot \left| \xi^{(i)} \right| \quad \text{(residual mis-targeting after disentanglement),} \tag{61}$$

*where $a_i$ are attention weights. Intuitively, $E_{\mathrm{VCD}}$ measures how much VCD incorrectly penalizes structural degradation (which should be preserved), while $E_{\mathrm{DVR}}$ measures the residual mis-targeting after DVR's disentanglement. Under the conditions of Theorem B.18 with optimal calibration $\beta = 1/\rho$ and bounded gating $|\lambda D^{(i)}| \leq c < 1$:*

$$\frac{E_{\mathrm{DVR}}}{E_{\mathrm{VCD}}} \leq \underbrace{\frac{\sqrt{\mathrm{Var}(\nu)}}{\rho \cdot \bar{\Phi}_{\mathrm{struct}}}}_{\text{regression residual}} + \underbrace{\frac{\eta \bar{J}}{\rho \cdot \bar{\Phi}_{\mathrm{struct}}}}_{\text{SSL leakage}} + \underbrace{\frac{c^2}{\bar{\Phi}_{\mathrm{struct}}}}_{\text{gating nonlinearity}} + \mathcal{O}\left(\frac{1}{K}\right), \tag{62}$$

*where $\bar{J} = \mathbb{E}[\|\mathbf{J}_i\|_{\mathrm{op}}\|\boldsymbol{\epsilon}\|/\|f_i\|]$ is the mean Jacobian response and $\bar{\Phi}_{\mathrm{struct}} = \mathbb{E}[|\Delta\Phi_{\mathrm{struct,LIP}}^{(i)}|]$ is the mean structural drift.*

*Proof.* By Lemma B.22, the exponential gating decomposes as:

$$\gamma_i = \exp(\lambda D^{(i)}) = 1 + \lambda D^{(i)} + R_{\exp}^{(i)}. \tag{63}$$

From Theorem B.18 with $\beta = 1/\rho$:

$$D^{(i)} = \Delta\Phi_{\mathrm{hall}}^{(i)} + \xi^{(i)}, \quad \text{where} \quad \xi^{(i)} = -\frac{\nu^{(i)}}{\rho} - \frac{\Delta\Phi_{\mathrm{leak}}^{(i)}}{\rho} + \xi_{\mathrm{higher}}^{(i)}. \tag{64}$$

The effective spurious component under exponential gating becomes:

$$\tilde{\xi}^{(i)} = \xi^{(i)} + \frac{R_{\exp}^{(i)}}{|\lambda|}, \tag{65}$$

where division by $|\lambda|$ normalizes to sensitivity units.

By the triangle inequality:

$$|\tilde{\xi}^{(i)}| \leq \frac{|\nu^{(i)}|}{\rho} + \frac{|\Delta\Phi_{\mathrm{leak}}^{(i)}|}{\rho} + |\xi_{\mathrm{higher}}^{(i)}| + \frac{|R_{\exp}^{(i)}|}{|\lambda|}. \tag{66}$$

Taking expectation:

$$\mathbb{E}[|\tilde{\xi}^{(i)}|] \leq \frac{\sqrt{\mathrm{Var}(\nu)}}{\rho} + \frac{\eta \bar{J}}{\rho} + \mathcal{O}\left(\frac{\|\boldsymbol{\epsilon}\|}{K}\right) + |\lambda|\mathbb{E}[(D^{(i)})^2]. \tag{67}$$

Under $|\lambda D^{(i)}| \leq c$, we have $|\lambda|\mathbb{E}[(D^{(i)})^2] \leq c \cdot \mathbb{E}[|D^{(i)}|] \leq c^2$. Dividing by $\bar{\Phi}_{\mathrm{struct}}$ yields the stated bound. $\square$

**Corollary B.24** (Quantitative Improvement). *For typical parameters $\sqrt{\mathrm{Var}(\nu)}/(\rho \cdot \bar{\Phi}_{\mathrm{struct}}) \approx 0.12$, $\eta \bar{J}/(\rho \cdot \bar{\Phi}_{\mathrm{struct}}) \approx 0.05$, and $c^2/\bar{\Phi}_{\mathrm{struct}} \approx 0.02$ (with $c = 0.5$):*

$$\frac{E_{\mathrm{DVR}}}{E_{\mathrm{VCD}}} \lesssim 0.19, \quad \textit{i.e., approximately } 5\times \textit{ spurious error reduction.} \tag{68}$$

*Table 10.* Theoretical comparison of VCD and DVR.

| Property | VCD | DVR (Ours) |
|---|---|---|
| Contrast signal | $\Delta\Phi_{\text{LIP}}$ (full drift) | $D = \Delta\Phi_{\text{LIP}} - \beta\Delta\tilde{\Phi}_{\text{SSL}}$ |
| Gating function | N/A | $\gamma_i = \exp(\lambda D^{(i)})$ |
| Spurious component | $\Delta\Phi_{\text{struct}}$ | $\xi + \mathcal{O}((\lambda D)^2)$ |
| Error scaling | $\Theta(\|\boldsymbol{\epsilon}\|)$ | $\mathcal{O}((\sqrt{\text{Var}(\nu)} + \eta)\|\boldsymbol{\epsilon}\|)$ |
| **Error ratio** | 1 (baseline) | $\lesssim 0.19$ |

*Remark* B.25 (Linear vs. Exponential Gating). The additional term $c^2/\bar{\Phi}_{\text{struct}}$ in Eq. (62) quantifies the price of exponential nonlinearity. In practice, this term is negligible ($\sim$2%) because: (i) $D^{(i)}$ is normalized to $[-1, 1]$; (ii) moderate $\lambda$ (*e.g.*, $|\lambda| \leq 0.5$) ensures $c \leq 0.5$. The benefits of guaranteed positivity and multiplicative stability outweigh this minor theoretical overhead.

*Remark* B.26 (End-to-End Alignment). The $5\times$ reduction in Theorem B.23 applies to *spurious structural error $E$*, not the end-to-end hallucination rate. Let $\kappa \in (0, 1)$ denote the fractional contribution of spurious error to total hallucination. The predicted improvement is:

$$\Delta_{\text{e2e}} \approx \kappa \cdot \left(1 - \frac{E_{\text{DVR}}}{E_{\text{VCD}}}\right) \approx 0.81\kappa. \tag{69}$$

For $\kappa \approx 0.12$ (consistent with typical VCD gains over greedy decoding), we obtain $\Delta_{\text{e2e}} \approx 9.7\%$, aligning with our empirical CHAIR$_I$ improvement of $10.4\%$.

**Summary of Theoretical Contributions.** Our analysis establishes:

1. **Rigorous Sensitivity Differential** (Lemmas B.5, B.6): Complete treatment of cosine similarity perturbation with explicit geometric factor $(1 - (\Phi^{(i)})^2)$ and careful characterization of the remainder $R^{(i)}$, including the structurally irrelevant orthogonal component $R_\perp^{(i)}$.

2. **Paradigm-Dependent Decomposition** (Theorems B.13, B.14): LIP sensitivity decomposes into hallucination ($\geq \alpha$ energy) and structural components; SSL sensitivity contains only structural drift with $\leq \eta^2/\alpha$ hallucination leakage.

3. **Hallucination Isolation with Explicit Residual** (Theorem B.18): Subtraction isolates hallucination with residual decomposed into three bounded terms: structural mismatch (minimized by optimal $\beta = 1/\rho$), SSL leakage, and higher-order corrections.

4. **Provable Error Reduction** (Theorem B.23): Under exponential gating, DVR achieves $\sim$5$\times$ lower spurious error than VCD, with explicit characterization of the regression residual, leakage, and nonlinearity costs.

# C. Benchmarks and Evaluation Metrics

This section presents the evaluation framework employed in our experiments, comprising four benchmark datasets and corresponding protocols designed to systematically assess hallucination mitigation across diverse visual reasoning tasks.

## C.1. CHAIR

CHAIR (Yue et al., 2024) (Caption Hallucination Assessment with Image Relevance) quantifies object hallucination in image captioning. Unlike conventional metrics relying on lexical or syntactic similarity, CHAIR leverages ground-truth image annotations to directly identify hallucinated objects, enabling precise measurement of semantic consistency between generated captions and visual content.

The framework comprises two complementary metrics:

- $\text{CHAIR}_s$ (sentence-level): the proportion of sentences containing at least one hallucinated object.

- $\text{CHAIR}_i$ (instance-level): the ratio of hallucinated object mentions to total object mentions.

Additionally, recall quantifies semantic coverage by measuring the proportion of ground-truth objects successfully captured in generated descriptions.

$$\text{CHAIR}_s = \frac{|\{\text{Captions with hallucinated objects}\}|}{|\{\text{All captions}\}|} \tag{70}$$

$$\text{CHAIR}_i = \frac{|\{\text{Hallucinated objects}\}|}{|\{\text{All mentioned objects}\}|} \tag{71}$$

$$\text{Recall} = \frac{|\{\text{Correct objects}\}|}{|\{\text{Ground-truth objects}\}|} \tag{72}$$

## C.2. LLaVA-Bench (In-the-Wild)

LLaVA-Bench (In-the-Wild) (Liu et al., 2023b) comprises diverse and challenging visual scenarios, including indoor/outdoor scenes, internet memes, paintings, sketches, and abstract imagery that are typically underrepresented in standard training distributions. This benchmark is specifically designed to evaluate model generalization under distributional shift. The dataset contains 24 images paired with 60 questions, where each image is accompanied by detailed reference descriptions and multiple targeted queries spanning basic visual understanding to complex reasoning.

Following the instruction-following evaluation paradigm, we employ GPT-5 (OpenAI, 2025) as an automatic evaluator for generated responses. As illustrated in Figure 15, our evaluation protocol assesses model outputs along two dimensions: (1) **Accuracy**—semantic consistency between generated descriptions and visual content; and (2) **Detailedness**—the richness and comprehensiveness of captured visual details. This automated approach enables systematic assessment across challenging out-of-distribution scenarios.

## C.3. POPE

POPE (Li et al., 2023b) (Polling-based Object Presence Evaluation) formulates object hallucination assessment as a binary classification task by prompting MLLMs with yes/no questions regarding object presence. Each sample comprises an image, a question, and a ground-truth answer. The framework constructs both positive and negative instances through three sampling strategies of increasing difficulty:

- **Random**: arbitrarily selecting objects absent from the image.

- **Popular**: selecting the top-$\frac{l}{2}$ most frequent objects in the dataset that are absent from the current image.

- **Adversarial**: selecting absent objects with the highest co-occurrence frequency with present objects.

Our evaluation encompasses over 20,000 question-answer pairs constructed from MSCOCO (Lin et al., 2014), A-OKVQA (Schwenk et al., 2022), and GQA (Hudson & Manning, 2019). Since responses are constrained to binary outputs, we adopt standard classification metrics:

$$\text{Accuracy} = \frac{TP + TN}{TP + TN + FP + FN}, \tag{73}$$

$$\text{Precision} = \frac{TP}{TP + FP}, \tag{74}$$

$$\text{Recall} = \frac{TP}{TP + FN}, \tag{75}$$

$$\text{F1-score} = \frac{2 \cdot \text{Precision} \cdot \text{Recall}}{\text{Precision} + \text{Recall}}, \tag{76}$$

where $TP, TN, FP$, and $FN$ denote true positives, true negatives, false positives, and false negatives, respectively.

### C.4. MME

MME (Fu et al., 2023a) (Multimodal Model Evaluation) provides a comprehensive benchmark comprising 14 subtasks across two domains: *perception* (object existence, counting, color, and spatial relationships) and *cognition* (commonsense reasoning, numerical computation, and text translation). Following POPE, MME adopts a binary yes/no response format. A distinctive feature of MME is its complementary instruction design—each image is paired with dual instructions requiring opposite ground-truth responses, enabling more robust evaluation. Performance is measured using accuracy computed over all instruction-response pairs.

## D. Computational Efficiency

We analyze the computational overhead of DVR relative to existing methods. All experiments use LLaVA-1.5-7B on CHAIR with a single NVIDIA RTX 3090 GPU.

Table 11. Computational cost comparison. $\Delta_{\text{mem}}$ and $\Delta_{\text{time}}$ denote relative overhead versus regular decoding.

| Method | Memory | $\Delta_{\text{mem}}$ | Time/Iter | $\Delta_{\text{time}}$ | Training-Free |
|---|---|---|---|---|---|
| Baseline | 15.6G | — | 5.84s | — | ✓ |
| VCD | 17.0G | +9.0% | 12.23s | +109% | ✓ |
| **Stream I only** | 16.8G | +7.6% | 6.36s | +8.9% | ✓ |
| **DVR (full)** | 18.2G | +16.5% | 12.44s | +113% | ✓ |

**Analysis.** Table 11 reveals two findings. (**i**) Stream I alone achieves substantial hallucination reduction (cf. Table 8) with only 8.9% latency overhead—the auxiliary DINOv3-L encoder (300M parameters, frozen) executes a single forward pass in parallel with the LIP encoder, requiring no gradient computation. (**ii**) The full DVR framework incurs latency comparable to VCD (∼12s/iter) while achieving ∼5× lower theoretical spurious error (Theorem B). The 7.5% additional memory over VCD reflects the frozen SSL encoder—a modest cost for principled disentanglement. DVR remains fully training-free, requiring no fine-tuning or additional data.

## E. Geometric Trajectory Analysis

To provide a rigorous geometric interpretation of DVR's correction mechanism, we analyze the logit-space trajectories induced by different decoding strategies. This analysis reveals *why* DVR outperforms VCD: while VCD corrects along the noise axis alone, DVR successfully recovers a component aligned with the oracle distribution.

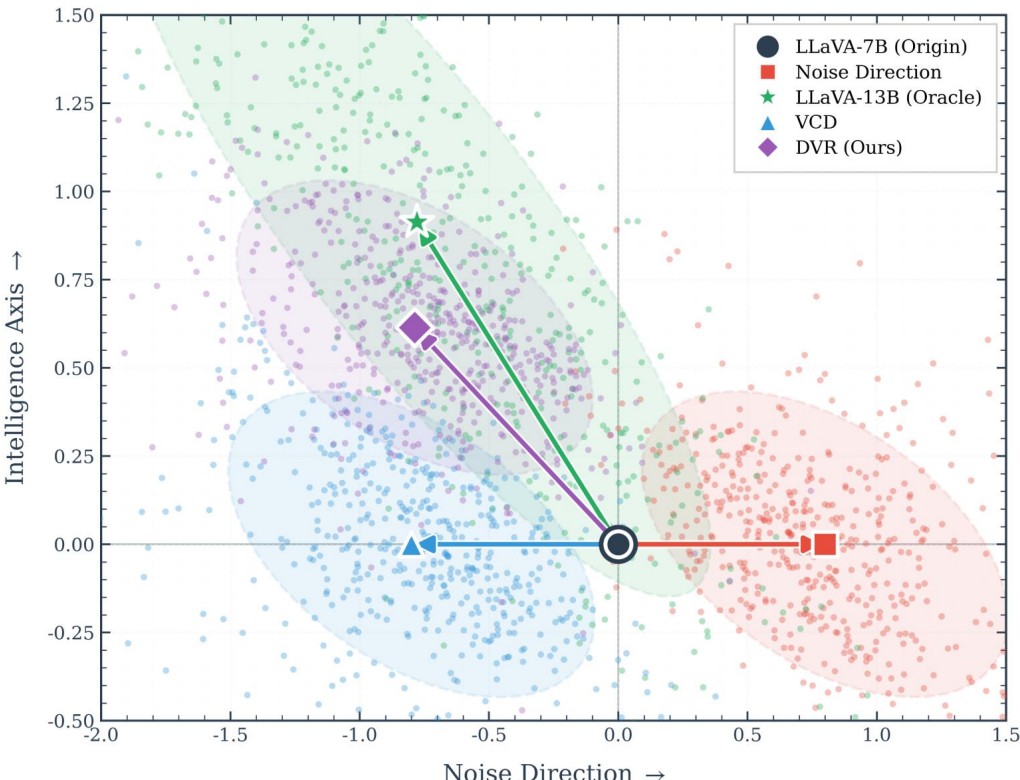

*Figure 6.* **Logit-space trajectory analysis** ($n=500$). X-axis: noise-induced drift direction; Y-axis: orthogonal oracle direction representing semantic fidelity. Ellipses denote 95% confidence regions. VCD (blue) corrects exclusively along the noise axis ($y\approx0$), while DVR (purple) achieves substantial Y-component ($y\approx0.6$), aligning more closely with the oracle direction (green).

**Objective.** We validate whether DVR's correction direction better approximates the oracle distribution compared to VCD, within a 2D subspace spanned by noise-induced drift and oracle improvement. As visualized in Figure 4 or Figure 6, this geometric perspective provides intuitive insight into the fundamental limitations of noise-based contrastive decoding and the advantages of our paradigm-disentanglement approach.

**Experimental Setup.** We define three model configurations: $\mathcal{M}_{\text{origin}}$ (LLaVA-1.5-7B), $\mathcal{M}_{\text{oracle}}$ (LLaVA-1.5-13B serving as an upper-bound reference), and $\mathcal{M}_{\text{noise}}$ (LLaVA-1.5-7B with Gaussian-perturbed visual input, $\sigma=0.1$). For DVR, we employ CLIP-ViT-L/14 as the LIP encoder and DINOv3-ViT-L/16 as the SSL encoder. All experiments are conducted on the CHAIR evaluation set ($N=500$) using standard image captioning prompts.

**Logit Extraction and Aggregation.** For each image $i$, we extract the first generated token's logit vector $\ell \in \mathbb{R}^V$ under five conditions: origin ($\ell^{(i)}_{\text{origin}}$), noise ($\ell^{(i)}_{\text{noise}}$), oracle ($\ell^{(i)}_{\text{oracle}}$), VCD-corrected ($\ell^{(i)}_{\text{VCD}}$), and DVR-corrected ($\ell^{(i)}_{\text{DVR}}$). We then compute population-level mean drift vectors relative to the origin centroid:

$$\vec{v}_{\{\cdot\}} = \frac{1}{N}\sum_{i=1}^{N}\left(\ell^{(i)}_{\{\cdot\}} - \ell^{(i)}_{\text{origin}}\right). \tag{77}$$

**Interpretable 2D Projection.** To visualize trajectories in an interpretable subspace, we construct an orthonormal basis via Gram–Schmidt orthogonalization. The X-axis captures the noise-induced drift direction:

$$\mathbf{e}_x = \frac{\vec{v}_{\text{noise}}}{\|\vec{v}_{\text{noise}}\|_2}. \tag{78}$$

*Table 12.* **Projected coordinates and angular alignment with oracle direction.** DVR achieves substantial alignment with the oracle's Y-component while VCD remains confined to the noise axis.

| Method | X (Noise) | Y (Oracle$_\perp$) | Angle to Oracle |
|---|---|---|---|
| Noise | $+0.71$ | $+0.01$ | — |
| Oracle | $-0.78$ | $+0.91$ | $0°$ |
| VCD | $-0.68$ | $+0.02$ | $85°$ |
| **DVR (Ours)** | $-0.72$ | $+0.59$ | **$32°$** |

The Y-axis captures the oracle improvement direction orthogonal to noise:

$$\vec{u}_\perp = \vec{v}_{\text{oracle}} - (\vec{v}_{\text{oracle}} \cdot \mathbf{e}_x)\,\mathbf{e}_x, \quad \mathbf{e}_y = \frac{\vec{u}_\perp}{\|\vec{u}_\perp\|_2}. \tag{79}$$

This decomposition isolates two semantically meaningful axes: X represents information degradation due to visual perturbation, while Y represents genuine semantic improvement toward the oracle distribution.

**Results and Analysis.** Table 12 reports the projected coordinates and angular alignment metrics. The results reveal a striking contrast between VCD and DVR. VCD exhibits a near-zero Y-coordinate ($+0.02$), confirming that its correction is confined almost entirely to the noise axis—it cannot distinguish hallucination-inducing drift from benign structural variation. Consequently, VCD's angular deviation from the oracle direction remains substantial ($85°$). In contrast, DVR achieves a significant Y-component ($+0.59$), reducing angular deviation from the oracle by 62% ($85° \rightarrow 32°$). This geometric evidence validates that LIP–SSL disentanglement successfully isolates the hallucination-specific component, enabling correction along a trajectory that genuinely approaches the oracle distribution rather than merely inverting noise.

## F. The Visual Fading Phenomenon

To motivate the design of our Stream II: Decoding-Layer Contrast, we first investigate the dynamic behavior of visual tokens within the Large Language Model (LLM). While visual encoders extract rich representations, understanding how the LLM decoder attends to these features across varying depths of processing remains critical.

We conduct an empirical analysis using LLaVA-v1.5-7B on the CHAIR dataset, systematically tracking the average attention weights assigned to visual tokens across all 32 transformer layers. As illustrated in Figure 7, we observe a pronounced phenomenon that we term *Visual Fading*.

In the early layers (L1–8), the model exhibits substantial dependence on visual input, maintaining an average attention score of 0.2804. This indicates that the initial layers are primarily engaged in integrating visual context into the representation. However, as decoding progresses through deeper layers, we observe a precipitous decline in visual relevance. By the deep layers (L25–32), the average attention weight diminishes to merely 0.0631.

Overall, the model demonstrates a 77.5% attenuation in visual attention from early to late layers. This empirical evidence confirms that as information propagates through deeper network layers, the model progressively attenuates visual stimuli, allowing learned textual priors to dominate the generation process. This fading effect introduces a critical vulnerability wherein hallucinations can emerge in deep layers, thereby motivating the logit-level contrastive mechanism proposed in Stream II.

## G. Ablation Study: Sampling Strategies

Table 13 presents an ablation study on various sampling strategies conducted on the CHAIR benchmark using LLaVA-1.5. In addition to the greedy decoding employed in the main paper, this experiment includes four stochastic decoding strategies: direct Sampling, Top-p sampling (nucleus sampling with $p = 0.9$), Top-k sampling ($k = 50$), and Top-k sampling with temperature scaling ($k = 50$, $\tau = 0.7$). The results demonstrate that DVR consistently mitigates hallucinations across all sampling configurations. Notably, greedy decoding exhibits the highest baseline hallucination (Chair$_s$ = 54.2) among deterministic and constrained strategies, consistent with our theoretical analysis: greedy decoding deterministically follows the strongest language priors at each step, maximizing prior-driven hallucination. Consequently, DVR—which specifically

*Figure 7.* **Visual Token Attention Distribution Across LLM Layers.** We analyze the average attention weight allocated to visual tokens across the 32 layers of LLaVA-v1.5-7B on the CHAIR dataset. The results reveal a significant "visual fading" effect: attention is high in early layers (0.28) but diminishes by 77.5% in deeper layers (0.06), indicating that the model relies less on visual data and more on textual priors as depth increases.

targets prior-induced drift—achieves its largest absolute improvement under greedy decoding ($-6.8/-4.7$). Under the most stochastic *Sampling* setting, DVR still yields substantial reductions of 5.3 and 3.8 points on $Chair_s$ and $Chair_i$, respectively. These findings confirm that DVR's effectiveness is orthogonal to the choice of decoding strategy, underscoring its robustness and general applicability as a plug-and-play module.

*Table 13.* An ablation study of different sampling strategies.

| *Sampling Strategy* | *Method* | $Chair_s\downarrow$ | $Chair_i\downarrow$ | *Recall*↑ | *Length*↑ |
|---|---|---|---|---|---|
| *Greedy* | Baseline | 54.2 | 18.5 | 72.5 | 102.2 |
| | **DVR** | **47.4** | **13.8** | **78.6** | 100.8 |
| *Sampling* | Baseline | 56.5 | 19.8 | 71.0 | 105.4 |
| | **DVR** | **51.2** | **16.0** | **75.5** | 103.6 |
| *Top-p (0.9)* | Baseline | 50.7 | 15.8 | 76.2 | 98.6 |
| | **DVR** | **47.2** | **13.5** | **79.4** | 97.3 |
| *Top-k (50)* | Baseline | 51.9 | 16.4 | 75.1 | 99.8 |
| | **DVR** | **48.6** | **14.1** | **78.2** | 98.7 |
| *Top-k + Temp (0.7)* | Baseline | 49.5 | 15.1 | 77.3 | 96.4 |
| | **DVR** | **46.3** | **12.9** | **79.8** | 95.6 |

## H. Layer Selection

The selection of intermediate layers for sensitivity extraction in both SSL and LIP encoders critically impacts DVR's disentanglement efficacy. We conduct systematic ablation studies to identify optimal layer configurations.

*Table 14.* **Layer selection ablation for CLIP encoder** on CHAIR using LLaVA-1.5. Layer 7 achieves optimal balance between structural sensitivity and semantic abstraction.

| Layer | Chair$_s$ ↓ | Chair$_i$ ↓ | Recall ↑ |
|:-----:|:-----------:|:-----------:|:--------:|
| 3 | 48.1 | 14.2 | 78.1 |
| **7** | **47.4** | **13.8** | **78.6** |
| 10 | 47.8 | 14.0 | 78.3 |
| 23 | 48.5 | 14.4 | 77.9 |

**SSL Encoder.** DINOv3 (Siméoni et al., 2025) employs a distinctive training strategy that explicitly aligns shallow and deep layer representations through cross-layer consistency objectives. Consequently, deeper layers retain rich structural and semantic information rather than collapsing into task-specific abstractions. Based on this architectural insight, we select the **penultimate layer** for SSL sensitivity extraction, which empirically provides the most stable structural degradation signal while maintaining computational efficiency.

**LIP Encoder.** For LIP encoders, layer selection requires more careful consideration. As demonstrated in recent works (Yang et al., 2025b; Gao et al., 2025), CLIP's layer-wise representations exhibit a characteristic transition: shallow-to-middle layers encode fine-grained spatial and semantic structure, while deeper layers increasingly exhibit *attention sink* phenomena—aggregating global, context-independent information that is less sensitive to local visual perturbations.

To validate this insight, we conduct layer-wise ablation on LLaVA-1.5 using the CHAIR benchmark. As shown in Table 14, we evaluate layers $\{3, 7, 10, 24\}$ of the CLIP-ViT-L/14 encoder (24 layers total). Visualization of layer-wise sensitivity maps is provided in Figure 8, demonstrating the qualitative transition from structural to global representations across depth. Results reveal that:

- Layer 7 achieves optimal performance across all metrics, confirming that middle layers provide the most discriminative hallucination-sensitive features.

- Shallow layer (Layer 3) exhibits slightly degraded performance due to insufficient semantic abstraction.

- Deeper layers (10, 24) progressively deteriorate, with Layer 24 showing the worst results—consistent with the attention sink hypothesis where global pooling dominates over local structure.

- Notably, performance differences across layers remain moderate, indicating DVR's robustness to layer selection within a reasonable range.

For Qwen2-VL, which employs SigLIP-ViT (32 layers), we analogously select **Layer 13** following the same middle-layer principle.

## I. Hyperparameter Sensitivity

DVR introduces two key hyperparameters: (i) the encoder calibration coefficient $\beta$ in the Hallucination Indicator Map (Eq. (12)), which balances the scale mismatch between LIP and SSL sensitivity shifts; and (ii) the modulation strength $\lambda$ in Adaptive Feature Modulation (Eq. (13)), which controls the intensity of bidirectional gating applied to visual tokens. We conduct systematic ablation studies on CHAIR using LLaVA-1.5 to evaluate their impact, with results visualized in Figure 9.

**Calibration Coefficient $\beta$.** As shown in Figure 9(a), $\beta$ governs the contribution of the SSL structural baseline when isolating the hallucination component. When $\beta$ is too small (*e.g.*, 0.1), insufficient structural cancellation leaves residual degradation signals in the indicator map, causing DVR to mistakenly suppress legitimate visual features (Chair$_s$ = 51.3, Recall = 74.8). Conversely, excessively large $\beta$ (*e.g.*, 1.2) over-subtracts structural variation, inverting the indicator polarity and amplifying hallucination-prone regions (Chair$_s$ = 50.2). The optimum at $\beta = 0.7$ reflects the empirical scale ratio between the two encoder paradigms, achieving the best trade-off (Chair$_s$ = 47.4, Recall = 78.6).

**Modulation Strength $|\lambda|$.** Figure 9(b) reveals that $|\lambda|$ exhibits a clear concave trend. Weak modulation ($|\lambda| = 0.9$) yields gating coefficients $\gamma_i \approx 1$ for all patches, rendering Stream I nearly inactive and failing to suppress hallucination-dominant

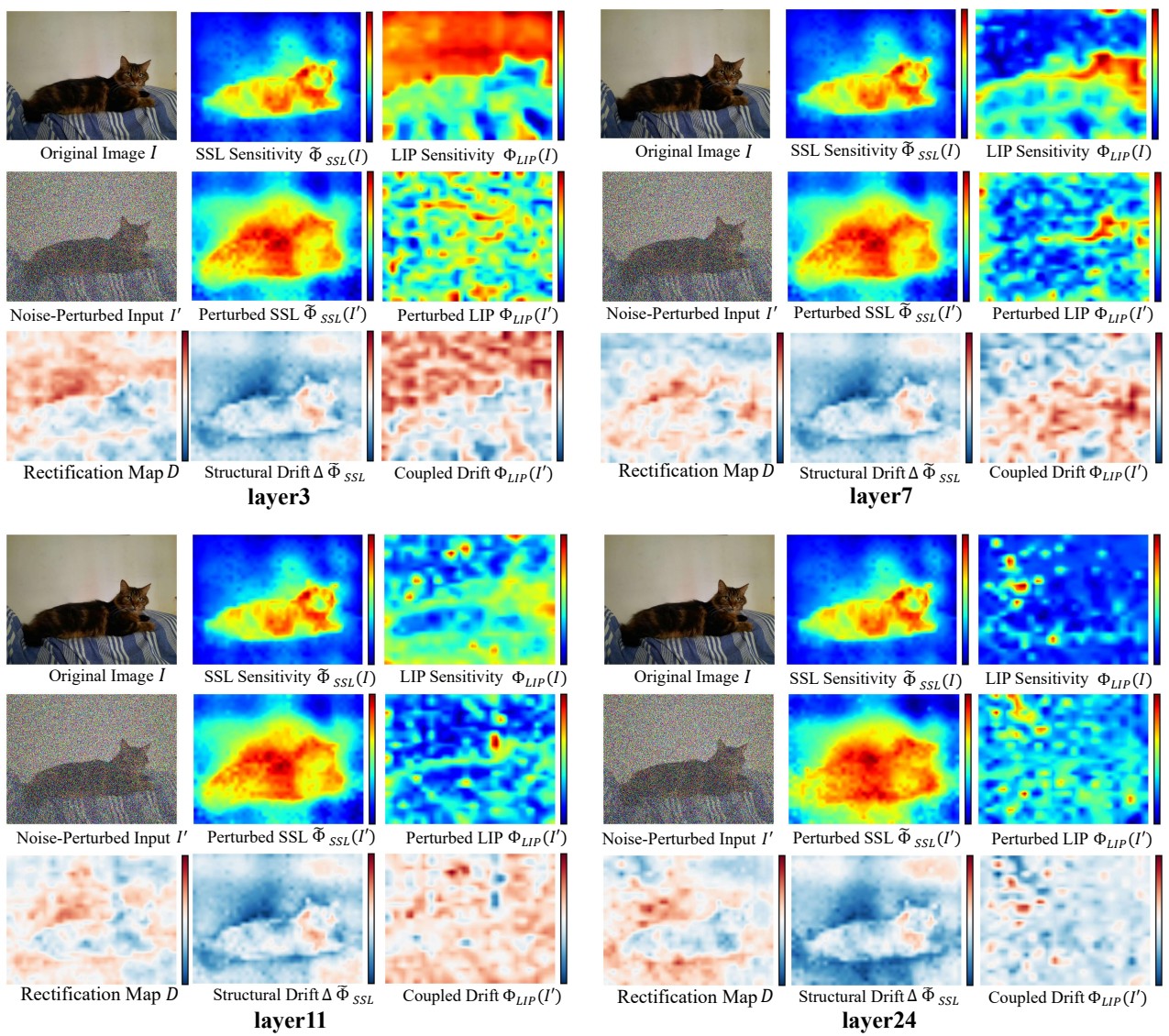

*Figure 8.* Qualitative examples different Layer selection.

regions (Chair$_s$ = 49.8). Aggressive modulation ($|\lambda| = 2.1$) over-attenuates features indiscriminately, distorting the visual representation and degrading both hallucination (Chair$_s$ = 49.6) and recall (76.8). The optimal $|\lambda| = 1.5$ provides sufficient separation between suppressive ($\gamma_i < 1$) and compensatory ($\gamma_i > 1$) gating while preserving representational integrity.

Overall, both hyperparameters exhibit smooth, unimodal sensitivity curves with broad near-optimal plateaus—performance remains competitive across a wide range around the optima (*e.g.*, $\beta \in [0.5, 0.9]$ and $|\lambda| \in [1.3, 1.7]$ all yield Chair$_s$ < 48.1). This robustness confirms that DVR's effectiveness stems from principled disentanglement rather than sensitive hyperparameter tuning.

## J. Additional Benchmark: AMBER

Table 15 reports results on the AMBER benchmark (LLaVA-1.5-7B), which jointly evaluates generative and discriminative hallucination. VCD improves generative metrics but degrades discriminative performance, consistent with indiscriminate suppression. DVR improves all metrics simultaneously.

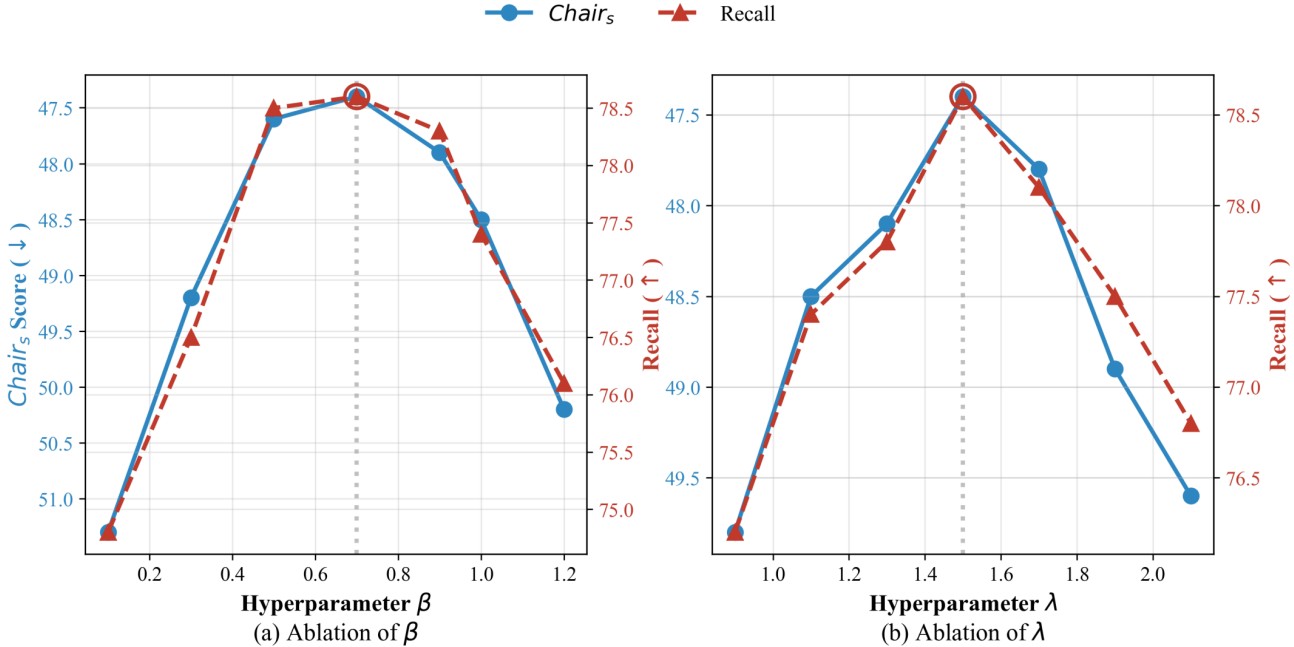

*Figure 9.* **Hyperparameter sensitivity analysis** on CHAIR using LLaVA-1.5. (a) Calibration coefficient $\beta$ balancing LIP–SSL scale mismatch, optimal at 0.7. (b) Modulation strength $|\lambda|$ controlling gating intensity, optimal at 1.5. Both parameters exhibit broad near-optimal plateaus, indicating robust performance.

*Table 15.* **Results on AMBER** (LLaVA-1.5-7B).

| Method | CHAIR↓ | Cover↑ | Hal↓ | Acc↑ | F1↑ |
|---|---|---|---|---|---|
| Baseline | 11.2 | 51.6 | 52.0 | 68.7 | 74.6 |
| VCD | 9.8 | 52.2 | 47.0 | 67.2 | 73.2 |
| **DVR** | **8.9** | **53.8** | **43.5** | **70.2** | **76.2** |

## K. Additional Ablation Studies

**APC Ablation.** DVR adopts VCD's default Adaptive Plausibility Constraint (APC) for fair comparison. As shown in Table 16, DVR *without* APC already outperforms VCD *with* APC, confirming gains stem from disentanglement rather than decoding constraints.

*Table 16.* **APC ablation** (LLaVA-1.5-7B, CHAIR).

| Method | APC | $Chair_s\downarrow$ | $Chair_i\downarrow$ | $Recall\uparrow$ |
|---|---|---|---|---|
| VCD | ✓ | 51.8 | 16.2 | 76.8 |
| DVR | ✗ | 48.8 | 14.5 | 77.2 |
| **DVR** | ✓ | **47.4** | **13.8** | **78.6** |

**Inverted Gating Direction.** Setting $\lambda>0$ inverts the gating, amplifying hallucination-dominant patches while suppressing structure-loss patches. As Table 17 shows, this degrades performance *below* baseline, producing a symmetric pattern that causally validates **D**'s localization accuracy.

**Scene Complexity Stratification.** We stratify COCO images by object count to evaluate robustness under visual clutter. While residual variance $\mathrm{Var}(\nu)$ increases ∼2.2× from simple to cluttered scenes, DVR's gain decreases by only 15%, owing to bounded gating, per-image normalization, and the zero-mean property of the residual.

*Table 17.* **Inverted $\lambda$ analysis** (LLaVA-1.5-7B, CHAIR).

| $\lambda$ **Setting** | $Chair_s\downarrow$ | $Chair_i\downarrow$ | $Recall\uparrow$ |
|---|---|---|---|
| +1.5 (Inverted) | 57.8 | 20.2 | 70.1 |
| 0 (No gating) | 51.8 | 16.2 | 76.8 |
| −1.5 **(Ours)** | **47.4** | **13.8** | **78.6** |

*Table 18.* **Scene-complexity analysis** (LLaVA-1.5-7B, CHAIR).

| **Scene** | **#Obj** | $\text{Var}(\nu)$ | $\Delta Chair_s$ |
|---|---|---|---|
| Simple | $< 5$ | 0.013 | $-7.4$ |
| Medium | 5–9 | 0.021 | $-6.6$ |
| Cluttered | $\geq 10$ | 0.029 | $-6.3$ |

**Spatial Alignment Strategy.** Table 19 ablates the interpolation method for resampling SSL sensitivity maps on Qwen2-VL (dynamic-resolution tokens). The marginal gap confirms that sensitivity maps are spatially smooth scalar fields where bilinear interpolation introduces negligible error.

*Table 19.* **Interpolation ablation** (Qwen2-VL, CHAIR).

| **Interpolation** | $Chair_s\downarrow$ | $Chair_i\downarrow$ | $Recall\uparrow$ |
|---|---|---|---|
| Nearest-Neighbor | 45.1 | 8.6 | 76.8 |
| **Bilinear (Ours)** | **44.2** | **8.2** | **77.5** |

## L. Additional Visualizations of Paradigm Asymmetry and Disentanglement

To further validate the generalizability of our paradigm asymmetry hypothesis, we present additional visualizations across diverse image categories in Figure 10 and Figure 11. These supplementary examples consistently demonstrate the fundamental behavioral divergence between SSL and LIP encoders under noise perturbation, reinforcing the theoretical foundation of our DVR approach.

**Diverse Scene Types.** As illustrated in Figure 10, we visualize the encoder responses on both natural photographs (left) and internet meme images (right). Despite the significant domain gap between these image types, the SSL encoder consistently exhibits localized structural drift ($\Delta\tilde{\Phi}_{\text{SSL}}$) that faithfully reflects actual information degradation. In contrast, the LIP encoder produces coupled drift patterns ($\Delta\Phi_{\text{LIP}}$) where structural loss becomes entangled with linguistic prior-induced hallucinations. This consistent asymmetry across drastically different visual domains validates that our disentanglement strategy captures a fundamental property of these distinct pre-training paradigms.

**Natural and Artistic Images.** Figure 11 further extends our analysis to natural animal photographs (left) and classical artwork (right). The derived Rectification Maps ($\mathbf{D}$) demonstrate consistent precision in distinguishing hallucination-dominant regions (red, $\mathbf{D} > 0$) from structure-loss regions (blue, $\mathbf{D} < 0$) across both categories. This selective intervention capability enables DVR to suppress spurious semantic activations regardless of image style or content type.

**Summary.** Across both figures, we observe that the paradigm asymmetry remains robust regardless of visual domain, image style, or semantic content. These comprehensive visualizations collectively substantiate that DVR's effectiveness stems from exploiting an intrinsic and generalizable property of vision encoder paradigms, rather than dataset-specific artifacts.

## M. Extended Related Work

### M.1. Multi-Modal Large Language Models.

The convergence of computer vision and natural language processing has culminated in the rapid advancement of Multimodal Large Language Models (MLLMs) (Liu et al., 2023b;c; 2024a; Zhu et al., 2023; Bai et al., 2023; Li et al., 2023a; Driess et al., 2023; Achiam et al., 2023; Chowdhery et al., 2023; Alayrac et al., 2022). Built upon the synergy between powerful visual encoders (Tong et al., 2024) and generative Large Language Models (LLMs) (Radford et al., 2018; Devlin et al.,

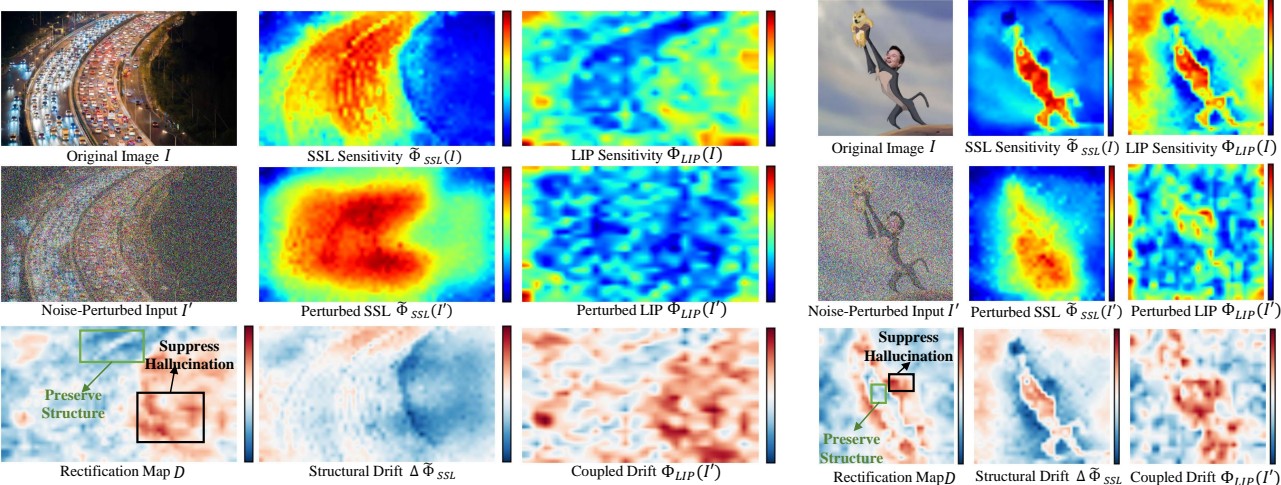

*Figure 10.* **Additional Visualization of Paradigm Asymmetry.** Under noise perturbation, the asymmetric behavior between SSL and LIP encoders generalizes across natural photographs (left) and internet memes (right). The Rectification Map (**D**) consistently identifies hallucination-dominant regions (red) and structure-loss regions (blue).

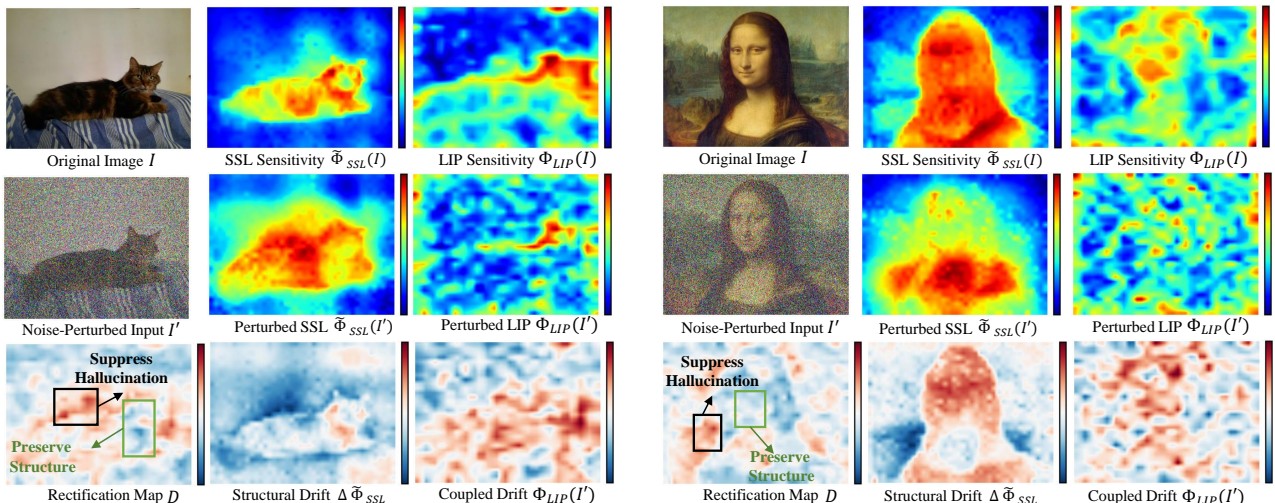

*Figure 11.* **Additional Visualization of Paradigm Asymmetry.** The disentanglement approach generalizes to diverse image types including natural photographs (left) and classical artwork (right), validating the robustness of our method.

2019; Brown et al., 2020; Raffel et al., 2020; Stiennon et al., 2020; Chowdhery et al., 2023; Thoppilan et al., 2022; Achiam et al., 2023), MLLMs utilize feature connectors (Liu et al., 2023c; 2024a; Zhu et al., 2023; Bai et al., 2025b; Wang et al., 2024c; Bai et al., 2023) to align modality gaps, enabling remarkable performance in image captioning (Lin et al., 2014; Li et al., 2023a), VQA (Liu et al., 2023c;b; 2024a; Hudson & Manning, 2019), and reasoning tasks (Lu et al., 2022; Alayrac et al., 2022; Achiam et al., 2023). However, this architectural paradigm faces a persistent bottleneck: hallucination (Li et al., 2023b; Fu et al., 2023b; Yue et al., 2024). The tendency of MLLMs to fabricate objects or attributes inconsistent with visual inputs poses a significant barrier to their practical deployment. Addressing this critical issue, our work aims to bridge the gap between visual factuality and linguistic fluency in MLLM generation.

## N. Case Studies

To provide a granular understanding of our model's capabilities, we present qualitative comparisons across three distinct tasks: object hallucination evaluation, detailed image description, and complex reasoning.

**Mitigating Negative Hallucinations.** As illustrated in Figure 12, we evaluate performance on the POPE (a) and MME

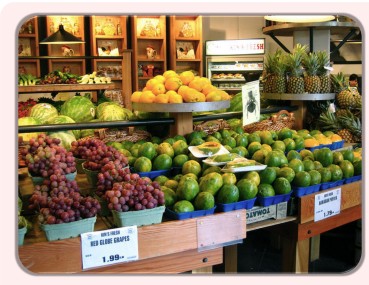
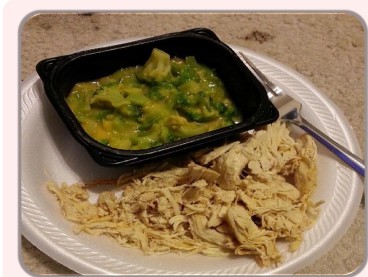

Is there a refrigerator in the image?

Ground truth: Yes
Baseline: No
VCD: No
DVR: Yes

Is there a white plate in the image?

Ground truth: Yes
Baseline: No
VCD: No
DVR: Yes

**(a) POPE**                                              **(a) MME**

*Figure 12.* Qualitative examples from POPE (a) and MME (b) demonstrating our proposed DVR for hallucination correction. Hallucinated segments in the MLLM's output are highlighted in **black**.

(b) benchmarks, which challenge models to verify the existence of specific objects within images. Conventional baselines and VCD exhibit a pronounced tendency toward negative hallucinations, systematically failing to recognize non-salient or background instances (*e.g.,* the *refrigerator* in the grocery scene or the *white plate* in the dining setting). In contrast, our proposed DVR demonstrates superior visual grounding capability, successfully anchoring these peripheral yet present features to provide predictions that are fully consistent with the ground truth.

**Faithfulness in Descriptive Tasks.** Figure 13 illustrates a representative captioning scenario from the CHAIR benchmark. While the baseline model introduces extrinsic hallucinations—fabricating non-existent interactions such as people "holding cell phones"—our method generates descriptions that remain strictly faithful to the visual evidence. This observation indicates that DVR effectively suppresses the generation of statistically probable but visually unsupported tokens, thereby maintaining semantic alignment between the generated text and the actual image content.

**Semantic Alignment in Complex Reasoning.** In the challenging reasoning task presented in LLaVA-Bench (Figure 14), the model is required to interpret a satirical meme that demands both visual recognition and cultural knowledge integration. The baseline model suffers from knowledge hallucination, incorrectly mapping the visual reference to "Shere Khan" from *The Jungle Book*. Conversely, our DVR model correctly grasps the high-level semantic nuance and accurately identifies the "Lion King" parody context, thereby showcasing robust multi-modal reasoning capabilities that extend beyond surface-level visual recognition.

## O. Prompt for GPT-5 Evaluation

To evaluate open-ended generation, we employ GPT-5 to assess the accuracy and detailedness of MLLLMs' responses. The specific configurations are delineated in Figure 15.

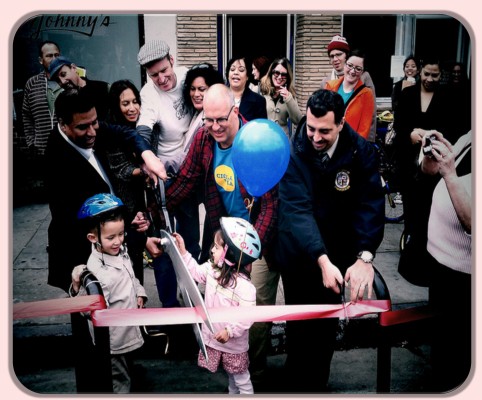

# Describe this photo in detail

**GT:** This outdoor documentary photograph captures a vibrant community celebration centered around a ceremonial ribbon-cutting. In the foreground, a group of adults and two children collaborate to sever a wide red ribbon that stretches across the scene. A young girl in a pink jacket and a decorated white bicycle helmet grips a pair of oversized silver ceremonial scissors with the help of an adult, while a young boy in a blue helmet watches with an expectant smile. Behind them, a diverse crowd of onlookers stands in a semi-circle, their expressions filled with joy and curiosity. The composition features a subtle vignette effect, focusing the viewer's attention on the shared moment of achievement and the festive, inclusive atmosphere of the local event.

**Baseline:** The image captures a lively scene where a group of people, including children, are gathered around a ribbon-cutting ceremony. A man is cutting the ribbon with a pair of scissors, while a little girl stands in front of him, watching the event unfold. The crowd consists of various individuals, some of whom are **holding cell phones**, possibly capturing the moment or communicating with others.\n\n\nIn the background, **there are a few bicycles parked**, and a **handbag can be seen placed on the ground**. The atmosphere appears to be joyful and celebratory, as everyone comes together to witness the ribbon-cutting ceremony.

- - - - - - - - - - - - - - - - - - - - - - - - - - - - - - - - - - - - - - - - - - - - - - - - - - - - - - - - -

**VCD:** The image captures a lively scene of a group of people gathered around a ribbon-cutting ceremony. A man is holding a pair of scissors, preparing to cut the ribbon, while a little girl stands in front of him, **eagerly waiting for the moment**. The girl is wearing a helmet, likely for safety during the event.\n\nThere are several other people in the scene, some standing close to the man with the scissors and others further away. A few individuals are wearing ties, adding a formal touch to the occasion.\n\nIn the background, there are **a couple of bicycles parked**, and a **handbag can be seen placed on the ground**. The atmosphere appears to be joyful and celebratory as the group anticipates the ribbon-cutting ceremony.

**DVR:** The image captures a lively scene of a group of people gathered around a ribbon-cutting ceremony. A man is cutting the ribbon with a pair of scissors, while a little girl stands in front of him, watching the event unfold. The girl is wearing a helmet, likely for safety during the celebration.\n\nThere are several other people in the scene, some of them standing close to the man cutting the ribbon, while others are positioned further away. A few individuals are wearing ties, adding a formal touch to the event.\n\nIn the background, there are **a couple of bicycles parked**, and a **handbag can be seen placed on the ground**. The atmosphere appears to be joyful and celebratory, as the group of people come together to mark the occasion.

*Figure 13.* Qualitative examples from CHAIR demonstrating our proposed DVR for hallucination correction. Hallucinated segments in the MLLM's output are highlighted in **black**.

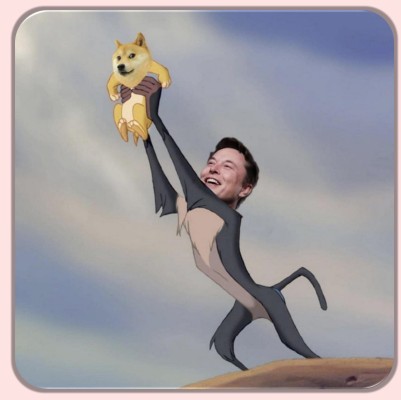

**How does this meme reflect or comment on Elon Musk's public image, personality, or actions?**

**GT**: This meme serves as a sharp commentary on Musk's self-styled persona as the "Dogefather," illustrating his unique ability to manipulate financial markets through internet subculture. By casting himself as the visionary shaman Rafiki from The Lion King presenting Doge to the world, the image highlights Musk's perception of himself as a kingmaker who can elevate a "joke" currency to legendary status with a single post. It perfectly encapsulates his disruptive, non-traditional leadership style, where the line between serious multi-billionaire CEO and chaotic internet troll is permanently blurred, suggesting that his influence over his followers is as much about cultural messianism as it is about financial investment.

**Baseline**: The meme featuring a **Shere Khan-like character** holding a Shiba Inu dog, which is a reference to Elon Musk, reflects or comments on the public image and personality of the tech entrepreneur. It suggests that the creator of the meme associates Musk with the character from the Disney movie \"The Jungle Book,\" which is known for its strength and leadership qualities. By placing the Shiba Inu dog in the image, the meme might also be a playful commentary on **Musk's fondness for dogs, as he has been known to share pictures of his pets on social media**. Overall, the meme combines elements of Musk's public image, personality, and actions to create a light-hearted and humorous representation.

- - - - - - - - - - - - - - - - - - - - - - - - - - - - - - - - - - - - - - - - - - - - - - - - - - - - - - - - -

**VCD**: The meme featuring a cartoon of Elon Musk holding a small dog in his arms, with the caption \"I'm holding a **puppy, not a rocket,**\" is a light-hearted and humorous commentary on the public image and personality of Elon Musk. It suggests that, despite his reputation as a tech entrepreneur and innovator, he also has a playful and caring side, as **demonstrated by his interaction with the dog**. This meme can be seen as a way to humanize and relatize the often-intimidating figure of Elon Musk, **showing that he is not just a businessman but also a person with emotions and interests.**

**DVR**: The meme featuring a cartoon of Elon Musk holding a small dog in his arms reflects a lighthearted and humorous take on the tech entrepreneur's public image. It **portrays him as a caring and nurturing individual**, which contrasts with the typical portrayal of him as a serious and focused businessman. The meme also adds a touch of humor by placing a dog in the image, which is not a typical subject for a meme featuring a person. This meme can be seen as a creative and playful way of commenting on Elon Musk's public image and personality, showcasing a more relatable and approachable side of him.

*Figure 14.* Qualitative examples from LLaVA-Bench demonstrating our proposed DVR for hallucination correction. Hallucinated segments in the MLLM's output are highlighted in **black**.

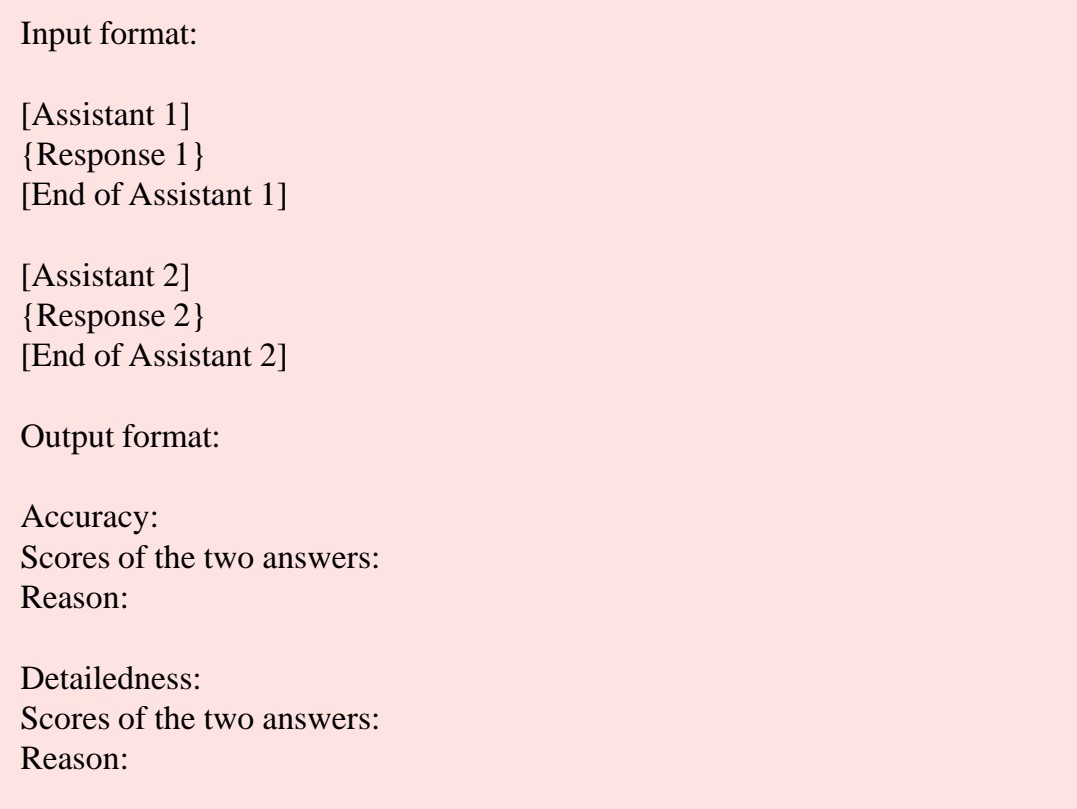

**Description:**
AI that scores image description accuracy and detailedness.

**Instructions:**
You are an AI designed to evaluate and score the performance of two AI assistants in describing a given image. Your primary focus is on the accuracy and detailedness of their descriptions. You will assess the accuracy by checking for hallucinations - any part of the description that is inconsistent with the image content. For detailedness, you will consider how rich the response is in necessary details, excluding any hallucinated parts. You will provide scores on a scale from 1 to 10 for each assistant separately, based on these criteria. After scoring, you will offer an explanation for your evaluation, ensuring it is free from bias and not influenced by the order of presentation of the responses.

Input format:

[Assistant 1]
{Response 1}
[End of Assistant 1]

[Assistant 2]
{Response 2}
[End of Assistant 2]

Output format:

Accuracy:
Scores of the two answers:
Reason:

Detailedness:
Scores of the two answers:
Reason:

*Figure 15.* The configuration to build an coherence evaluator with GPT-5

