# OpenReview forum: "Beyond Blind Noising: Disentangled Visual Rectification for Hallucination Mitigation in MLLMs"
_ICML.cc/2026/Conference — ICML 2026 regular_

### Official Review · Reviewer_kXzC · 2026-02-23

**Soundness:** 3
**Presentation:** 3
**Significance:** 3
**Originality:** 3
**Overall Recommendation:** 4
**Confidence:** 3

**Summary:**

This paper argues that when Multimodal Large Language Models (MLLMs) employ the traditional contrastive decoding method, VCD, they suffer from a loss of effective semantics. The authors attribute this to (i) structural degradation from corrupted visual information and (ii) hallucination induction from linguistic prior activation. To address this issue, the paper proposes Disentangled Visual Rectification (DVR), which leverages Self-Supervised Learning (SSL) encoders to perform visual-layer rectification and decoding-layer contrast on purified representations.

**Compliance With Llm Reviewing Policy:**

Affirmed.

**Final Justification:**

The authors address most of my concerns in the rebuttal. The added comparisons with recent training-free baselines and the new results on more recent architectures make the empirical case substantially stronger. My main remaining concern is practical efficiency. Although the rebuttal shows that the extra DINOv3 encoder adds only modest overhead relative to VCD, the full DVR method still remains substantially slower than standard decoding, which may limit practical deployment. Overall, the paper is strengthened by the rebuttal, and I maintain my recommendation.

**Key Questions For Authors:**

The empirical evaluation lacks comprehensive comparisons with a broader range of recent training-free algorithms. To clearly establish the specific advantages and superiority of the proposed approach over existing methods, the paper needs to include these baselines. Crucially, the manuscript misses a detailed comparison of practical deployment metrics (e.g. differences in inference time and computational overhead) against these existing training-free methods, making it difficult to fully assess the actual trade-offs of the proposed algorithm.

It remains unclear whether the specific hallucination phenomenon analyzed in this work still persists in more recent, state-of-the-art architectures, such as LLaVA-NeXT and InternVL-3. The manuscript would be significantly strengthened by providing empirical evidence on these newer models. Furthermore, the paper lacks an adaptive hyperparameter selection strategy, leaving it ambiguous how to systematically tune these parameters when applying the proposed method to different model architectures.

**Limitations:**

yes

**Strengths And Weaknesses:**

Strengths:

The paper provides a strong theoretical analysis revealing that the noise-induced drift in Language-Image Pretrained (LIP) encoders is fundamentally a coupled vector entangling. This insightful analysis convincingly demonstrates the inherent limitations and unreasonableness of standard Visual Contrastive Decoding (VCD).

By systematically comparing the performance of SSL encoders and LIP encoders under noisy conditions, the authors propose an innovative Disentangled Visual Rectification (DVR) approach. This method effectively disentangles the information flow to successfully retain the valid semantic components, which are then elegantly leveraged for robust contrastive decoding.

Weakness:

Although the paper successfully optimizes the VCD algorithm, contrastive decoding inherently suffers from slower inference speeds. Furthermore, the integration of the DINOv3 encoder inevitably leads to increased VRAM consumption during inference, which could limit the method's practical deployment and scalability.

While the proposed Disentangled Visual Rectification (DVR) method demonstrates superior hallucination suppression compared to standard contrastive decoding, the manuscript lacks empirical comparisons with other recent, state-of-the-art training-free algorithms. To properly benchmark the approach, the authors should include comparisons with methods such as:

[MemVR] Look Twice Before You Answer: Memory-Space Visual Retracing for Hallucination Mitigation. (ICML 2025)

[EAH] Seeing Clearly by Layer Two. (EMNLP 2025)

[Vissink] See What You Are Told: Visual Attention Sink in Large Multimodal Models. (ICLR 2025)

[VHR] Cracking the Code of Hallucination in LVLMs with Vision-aware Head Divergence. (ACL 2025)

The experimental evaluation is currently restricted to LLaVA-1.5 and Qwen2-VL. To substantially strengthen the paper's claims regarding generalizability and robustness, the authors should expand their evaluation to include more recent architectures introduced over the past two years, as well as models of varying parameter sizes.

---

> ### Author Rebuttal · Authors · 2026-03-30
>
> We thank Reviewer kXzC for recognizing our theoretical analysis and disentanglement approach. We address each concern below.
>
> ---
> ### **Q1: Inference speed and VRAM overhead of contrastive decoding + DINOv3.**
>
> **A1:**  We have expanded our efficiency analysis (Appendix D) to include all baselines evaluated under identical hardware conditions (LLaVA-1.5-7B, single RTX 3090, evaluating CHAIR):
>
> **Table R3: Computational cost comparison**
>
> |Method|Memory|Δmem|Time/Iter|Δtime|
> |-|-|-|-|-|
> |Baseline|15.6G|—|5.84s|—|
> |OPERA|23.4G|+50.0%|41.62s|+613%|
> |ICD|15.8G|+1.3%|6.21s|+6.3%|
> |VCD|17.0G|+9.0%|12.23s|+109%|
> |ONLY|15.8G|+1.3%|6.05s|+3.6%|
> |**DVR-Stream I**|**16.8G**|**+7.6%**|**6.36s**|**+8.9%**|
> |**DVR (full)**|**18.2G**|**+16.5%**|**12.44s**|**+113%**|
>
>
> In summary: DVR (full) adds only +0.21s and +1.2G over VCD, as the frozen DINOv3-L (300M params) is lightweight relative to the LLM passes that dominate cost. For latency-sensitive settings, **DVR-Stream I alone** outperforms VCD at +8.9% latency with 1 LLM pass.
>
> ---
> ###  **Q2: Missing comparisons with recent training-free methods (MemVR, EAH, Vissink, VHR).**
>
>
> **A2:** Thank you for pointing out these important recent works.
>
> We reproduced all four methods under our unified evaluation protocol (LLaVA-1.5-7B, 500 COCO images, identical prompts/sampling/max-length; Vissink re-implemented from the paper):
>
> |Method|CHAIR$_S$↓|CHAIR$_I$↓|Recall↑|POPE Acc↑|POPE F1↑|
> |-|-|-|-|-|-|
> |Baseline|54.2|18.5|72.5|83.2|83.4|
> |VCD|51.8|16.2|76.8|87.0|87.6|
> |Vissink|52.4|16.8|74.0|84.8|84.6|
> |EAH|48.0|14.8|74.5|84.2|84.0|
> |VHR|49.2|15.4|75.0|84.5|84.3|
> |MemVR|49.6|14.8|76.2|88.5|87.3|
> |**DVR**|**47.4**|**13.8**|**78.6**|**89.5**|**90.0**|
>
> **DVR achieves the best performance across all five metrics**, with notable margins on POPE F1 (+2.7 over MemVR) and Recall (+2.4). EAH/VHR show limited POPE improvement (+1.0–1.3 Acc), consistent with their original papers—they primarily reduce open-ended hallucination but struggle with discriminative judgments. DVR achieves +6.3 POPE Acc.
>
> **Fundamental distinction:** To clarify the fundamental differences, we contrast DVR with existing mechanisms:
>
> **recent Methods (Intervention During Generation)**: MemVR (re-emphasizes visual tokens), EAH (amplifies deep attention sinks), VHR (suppresses high-risk tokens), and Vissink (reallocates attention sinks) all manipulate attention/activations during the LLM's decoding phase to patch downstream symptoms.
>
> **DVR (Intervention Before Generation)**: In stark contrast, DVR leverages self-supervised encoders to rectify representations before they enter the LLM, preventing hallucinations at their source.
>
> This early intervention explains DVR's unique ability to simultaneously improve precision (lower CHAIR) and completeness (higher Recall). We will add this detailed comparison to the final version.
>
> ---
> ### **Q3: Evaluation on more recent and diverse architectures.**
>
> **A3:**  We extended evaluation to **Qwen3-VL-4B/8B (2025)** and the **AMBER benchmark**.
>
> **Table R1: Qwen3-VL on CHAIR**
>
> |Model|$CHAIR_S$↓|$CHAIR_I$↓|Recall↑|
> |--|-|-|-|
> |**Qwen3-VL-4B**||||
> |Baseline|26.0|9.4|49.2|
> |VCD|25.8|9.4|49.7|
> |**DVR**|**22.5**|**7.5**|**52.5**|
> |**Qwen3-VL-8B**||||
> |Baseline|21.0|7.8|46.1|
> |VCD|23.7|8.7|48.7|
> |**DVR**|**19.0**|**6.8**|**51.0**|
>
> **Table R2: LLaVA-1.5-7B on AMBER**
>
> |Method|CHAIR↓|Cover↑|Hal↓|Acc↑|F1↑|
> |-|-|-|-|-|-|
> |Baseline|11.2|51.6|52.0|68.7|74.6|
> |VCD|9.8|52.2|47.0|67.2|73.2|
> |**DVR**|**8.9**|**53.8**|**43.5**|**70.2**|**76.2**|
>
> The key finding: on Qwen3-VL-8B, VCD *worsens* hallucination (CHAIR$_S$ 21.0→23.7) while DVR improves (19.0). The hallucination phenomenon persists in modern architectures because their LIP encoders (SigLIP) retain the same image-text contrastive objective, making the structural-hallucination coupling DVR targets fundamentally persistent. Consistent improvements across 4 architectures (LLaVA-1.5, Qwen2-VL, Qwen3-VL-4B/8B) and 5 benchmarks confirm broad applicability.
>
> ---
> ### **Q4: Adaptive hyperparameter selection across architectures.**
>
> **A4:** We use **architecture-wise stable defaults** grounded in theory, not instance-wise tuning:
>
> - **$\beta$=0.7** (Eq. 12): Theory gives $\beta^*=1/\rho$ (Remark B.19). In practice, a **fixed default $\beta$=0.7** works robustly across all tested architectures. Fig. 9(a) confirms a broad near-optimal plateau (CHAIRS < 48.1 for $\beta$∈[0.5, 0.9]), demonstrating low sensitivity.
> - **$|\lambda|$=1.5** (Eq. 13): Fig. 9(b) shows a similarly broad plateau ($|\lambda| \in [1.3, 1.7]$). Appendix H–I further demonstrate stability across layer choices and hyperparameter ranges.
> - **$\alpha$=1** (Eq. 14): Inherited from VCD's standard setting.
>
> **Cross-architecture transfer:** The same defaults ($\beta$=0.7, $\lambda$=−1.5, $\alpha$=1) transfer from LLaVA-1.5 → Qwen2-VL → Qwen3-VL without any retuning, empirically validating cross-architecture robustness. We will make this practical guidance explicit in the final version.

---

> > ### Author Rebuttal · Reviewer_kXzC · 2026-04-03
> >
> > Thanks for the detailed response. I will maintain my positive score 4.

---

> > > ### Author Response · Authors · 2026-04-03
> > >
> > > Thank you very much for your careful review and encouraging feedback. We sincerely appreciate the time and effort you devoted to evaluating our paper and responses. We are glad that our clarifications have addressed your concerns. In the revised version, we will further incorporate the relevant clarifications to improve the presentation and overall clarity of the paper. We again thank you for your thoughtful comments and valuable suggestions.

---

### Official Review · Reviewer_3PJ6 · 2026-03-12

**Soundness:** 3
**Presentation:** 3
**Significance:** 2
**Originality:** 3
**Overall Recommendation:** 4
**Confidence:** 4

**Summary:**

This paper proposes Disentangled Visual Rectification (DVR), a training-free, dual-stream inference framework to mitigate hallucinations in multimodal LLMs. The key idea is to disentangle noise-induced feature drift into structural degradation and hallucination-inducing components by contrasting a Language-Image Pretrained (LIP) encoder with a Self-Supervised Learning (SSL) encoder. The authors provide geometric motivation, a theoretical error-reduction claim, and empirical results showing improvements over several training-free baselines on CHAIR, POPE, MME, and LLaVA-Bench.

**Compliance With Llm Reviewing Policy:**

Affirmed.

**Final Justification:**

They solve Inference cost.

**Key Questions For Authors:**

please see weaknesses

**Limitations:**

yes

**Strengths And Weaknesses:**

Strengths
- The core idea of leveraging the difference in inductive biases between a LIP encoder and a SSL encoder is insightful. By treating the SSL encoder as a "structural-only" probe, the paper introduces a principled geometric way to isolate and subtract the spurious, text-prior-induced drift that causes hallucinations.
- The framework logically divides the hallucination mitigation workload into two complementary mechanisms. The spatial visual-token rectification effectively addresses input-level corruption, while the subsequent contrastive decoding handles visual fading during text generation. This division of labor is both intuitive and well-articulated.

Weaknesses
- The paper's empirical claims rely entirely on 2023/2024-era models. In 2026, MLLM architectures have undergone drastic evolution, heavily utilizing dynamic resolution and AnyRes encoding, which yield vastly improved baseline hallucination profiles (e.g., Qwen3-VL). This misalignment makes it exceedingly difficult to assess whether DVR is a genuinely effective solution for modern multimodal hallucinations, or merely a band-aid for deprecated architectures.
- Although DVR is presented as a training-free method, its architectural design plus dual decoding—implies an heavy inference cost. It is necessary to see a comparison of end-to-end latency and peak memory overhead (VRAM) against listed baselines in Figure.1 under identical hardware configurations.

---

> ### Author Rebuttal · Authors · 2026-03-30
>
> We thank the reviewer for recognizing our principled geometric insight and dual-stream design. We address both concerns below.
>
> ---
>
> ###  **Q1: Evaluation limited to 2023/2024-era models; unclear effectiveness on modern architectures.**
>
> **A1:** We extended evaluation to **Qwen3-VL-4B/8B (2025)** and the **AMBER benchmark**.
>
> **Table R1: Qwen3-VL on CHAIR**
>
> |Model|$CHAIR_S$↓|$CHAIR_I$↓|Recall↑|
> |--|-|-|-|
> |**Qwen3-VL-4B**||||
> |Baseline|26.0|9.4|49.2|
> |VCD|25.8|9.4|49.7|
> |**DVR**|**22.5**|**7.5**|**52.5**|
> |**Qwen3-VL-8B**||||
> |Baseline|21.0|7.8|46.1|
> |VCD|23.7|8.7|48.7|
> |**DVR**|**19.0**|**6.8**|**51.0**|
>
> As shown, DVR achieves robust and consistent gains across Qwen3-VL-4B, Qwen3-VL-8B. Crucially, DVR provides meaningful improvements even when the baseline hallucination rate is already low (e.g., 21.0→19.0 on Qwen3-VL-8B). This demonstrates that DVR is not merely a band-aid for weaker models, but rather addresses a fundamental property of the LIP training paradigm.
>
>
> **Table R2: LLaVA-1.5-7B on AMBER**
>
> |Method|CHAIR↓|Cover↑|Hal↓|Acc↑|F1↑|
> |-|-|-|-|-|-|
> |Baseline|11.2|51.6|52.0|68.7|74.6|
> |VCD|9.8|52.2|47.0|67.2|73.2|
> |**DVR**|**8.9**|**53.8**|**43.5**|**70.2**|**76.2**|
>
>
> On AMBER, VCD degrades discriminative performance (Acc 68.7→67.2), while DVR improves both generative and discriminative metrics (Acc 70.2, F1 76.2).
>
> These results  demonstrates that DVR is not merely a band-aid for weaker models, but rather addresses a fundamental property of the LIP training paradigm.
>
> **Why DVR generalizes to dynamic/AnyRes architectures:** DVR's SSL encoder operates on the raw image—independent of the MLLM's internal resolution strategy. The sensitivity map alignment (Eq. 8–9) operates in continuous coordinates $\Omega=[0,1]^2$, naturally accommodating dynamic/AnyRes encoding. More fundamentally, even cutting-edge models (Qwen3-VL) employ LIP-based encoders (SigLIP) with the same image-text contrastive objective, so the structural-hallucination coupling DVR targets persists. The consistent improvements across 3 architectures and 5 benchmarks empirically confirm this.
>
> We will include Qwen3-VL and AMBER results in the main table of the final version.
>
> ---
> ### **Q2: Inference cost—latency and memory comparison with baselines.**
>
> **A2:** We deeply appreciate the reviewer highlighting this critical aspect for practical deployment. We have extended our efficiency analysis (Appendix D) to include **all baselines from Table. 1** under identical hardware (LLaVA-1.5-7B, single RTX 3090, CHAIR):
>
> **Table R3: Comprehensive computational cost comparison**
>
> |Method|Memory|Δmem|Time/Iter|Δtime|
> |-|-|-|-|-|
> |Baseline|15.6G|—|5.84s|—|
> |OPERA|23.4G|+50.0%|41.62s|+613%|
> |ICD|15.8G|+1.3%|6.21s|+6.3%|
> |VCD|17.0G|+9.0%|12.23s|+109%|
> |ONLY|15.8G|+1.3%|6.05s|+3.6%|
> |**DVR-Stream I**|**16.8G**|**+7.6%**|**6.36s**|**+8.9%**|
> |**DVR (full)**|**18.2G**|**+16.5%**|**12.44s**|**+113%**|
>
> **Key findings:**
> 1. **DVR (full) vs. VCD:** DVR introduces a nearly identical latency overhead compared to VCD (**+113% vs. +109%**), requiring only +7.5% additional memory over VCD to accommodate the frozen SSL encoder. This marginal overhead yields substantial performance gains ($\text{CHAIR}_S$ 47.4 vs. 51.8), making the cost-performance tradeoff highly favorable.
> 2. **DVR-Stream I as a practical lightweight option:** For latency-sensitive deployments, Stream I alone achieves $\text{CHAIR}_S$ 49.2 (vs. baseline 54.2, VCD 51.8) with merely **+8.9% latency and only 1 LLM pass**—faster than VCD while delivering stronger hallucination reduction. This makes DVR deployable even under strict latency constraints.
> 3. **Comparison with heavy methods:** While OPERA incurs a massive +613% latency overhead, DVR achieves superior or comparable hallucination reduction at a dramatically lower computational cost.
>
> We will promote this comprehensive table to the main paper in the final version to ensure the cost-performance tradeoff is transparently communicated. We believe these results solidify DVR as a principled and practical solution with highly favorable efficiency characteristics. Thank you for prompting this important analysis.

---

> > ### Author Rebuttal · Reviewer_3PJ6 · 2026-04-06
> >
> > Thank you for the response. I will increase score to 4.

---

> > > ### Author Response · Authors · 2026-04-07
> > >
> > > Thank you sincerely for taking the time to review our rebuttal and for adjusting your score. We truly appreciate it.
> > >
> > > Your two concerns — evaluation on modern architectures and inference cost analysis — were extremely constructive and directly strengthened our paper. The Qwen3-VL experiments and the comprehensive efficiency comparison table (Table R3) will both be prominently included in the final version, and we believe they make the paper significantly more complete and convincing.
> > >
> > > We are grateful for your recognition of our geometric insight and dual-stream design from the initial review, and for your continued engagement throughout this process. Your feedback has been invaluable in improving this work.

---

### Official Review · Reviewer_1wPA · 2026-03-13

**Soundness:** 3
**Presentation:** 3
**Significance:** 3
**Originality:** 3
**Overall Recommendation:** 3
**Confidence:** 5

**Summary:**

This paper starts with obtaining a patch level sensitivity map by estimating directional alignment of two different encoders, and using it to generate a rectified map. The logit from rectified map is contrasted with the logit from complete random noise to remove the language prior. The experiments are performed with 2 models on 4 benchmarks, and analyses are performed by ablating components, logit trajectories, and scale difference.

**Compliance With Llm Reviewing Policy:**

Affirmed.

**Key Questions For Authors:**

- Authors have mentioned that encoders “complete” degraded representations in noisy visual inputs. How can we validate this, and how does it induce hallucinations?
- Does the method consider adaptive plausibility constraint (APC) in the algorithm?
- How would the model behave if $\gamma$ is inversely signed?

**Limitations:**

Please refer to weakness and question sections.

**Strengths And Weaknesses:**

Strengths
- The idea of generating the perturbation based on the feature space is interesting.
- Logit space trajectory analysis and experimental results well supports the main idea of the paper

Weaknesses
There are two main concerns: experiment scale and latency.
- The experiments are performed with only LLaVA-1.5 and Qwen2-VL are considered outdated. It is desired to include more contemporary models in the main experiments. Also, it is recommended to test more benchmark.
- Results should be averaged with standard deviation across multiple runs [1,2,3]. The decoding based methods are implemented based on multinomial sampling, which inherits high randomness.
- Requirement of minimum 4 forward pass is a concern. Contrasting based decoding methods require multiple forward passes, therefore introduce higher computational overhead. Latency analysis should be performed to balance the tradeoff.

[1] Leng et al. Mitigating object hallucinations in large vision-language models through visual contrastive decoding. CVPR 2024

[2] Chen et al. HALC Object hallucination reduction via adaptive focal-contrast decoding. ICML 2024

[3] Im et al. Self-augmented visual contrastive decoding. ICLR 2026

---

> ### Author Rebuttal · Authors · 2026-03-30
>
> We thank Reviewer 1wPA for recognizing the novelty of our approach and the value of our trajectory analysis. We address each concern.
>
> ### **Q1: Experiments on more contemporary models and benchmarks.**
>
> **A1:** We extended evaluation to **Qwen3-VL-4B/8B (2025)** and the **AMBER benchmark**.
>
> **Table R1: Qwen3-VL on CHAIR**
>
> |Model|$CHAIR_S$↓|$CHAIR_I$↓|Recall↑|
> |--|-|-|-|
> |**Qwen3-VL-4B**||||
> |Baseline|26.0|9.4|49.2|
> |VCD|25.8|9.4|49.7|
> |**DVR**|**22.5**|**7.5**|**52.5**|
> |**Qwen3-VL-8B**||||
> |Baseline|21.0|7.8|46.1|
> |VCD|23.7|8.7|48.7|
> |**DVR**|**19.0**|**6.8**|**51.0**|
>
> **Critical finding:** On Qwen3-VL-8B, VCD *worsens* performance (CHAIR$_S$ 21.0→23.7), while DVR still improves (21.0→19.0). This directly validates our theory (§3.1): as baseline hallucination decreases, structural degradation dominates VCD's correction, causing it to suppress valid features. DVR's disentanglement avoids this failure mode. (Note: Lower absolute Recall reflects shorter generation; max_tokens=128 vs. 512(original paper). Relative DVR gains remain consistent.)
>
> **Table R2: LLaVA-1.5-7B on AMBER**
>
> |Method|CHAIR↓|Cover↑|Hal↓|Acc↑|F1↑|
> |-|-|-|-|-|-|
> |Baseline|11.2|51.6|52.0|68.7|74.6|
> |VCD|9.8|52.2|47.0|67.2|73.2|
> |**DVR**|**8.9**|**53.8**|**43.5**|**70.2**|**76.2**|
>
> VCD degrades discriminative performance on AMBER (Acc 68.7→67.2) due to indiscriminate suppression. DVR improves both generative and discriminative metrics.
>
> ### **Q2: Standard deviation across multiple runs.**
>
>
> **A2:** We re-ran sampling-based experiments with 3 seeds (LLaVA-1.5-7B, CHAIR):
>
> |**Method**|**CHAIRS↓**|**CHAIRI↓**|**Recall↑**|
> |-|-|-|-|
> |Baseline|54.3$\pm$1.3|18.4$\pm$0.6|72.5$\pm$0.7|
> |VCD|52.0$\pm$1.5|16.2$\pm$0.5|76.5$\pm$0.8|
> |**DVR**|**48.1$\pm$1.1**|**14.1$\pm$0.5**|**78.0$\pm$0.7**|
>
> DVR shows lower variability (SD: 1.1 vs. 1.5) and strictly non-overlapping bounds ([47.0, 49.2] vs. [50.5, 53.5]) compared to VCD, as Stream I stabilizes pre-decoding representations. All SDs will be reported. We have already cited HALC[2] and VCD[1], and we will cite Self-Aug [3] and add a comprehensive discussion in the final version.
>
> ### **Q3: Requirement of 4 forward passes and latency.**
>
> **A3:** DVR requires 4 **vision-encoder** passes (2×LIP + 2×SSL) and 2 **LLM passes**. Since LLM passes dominate latency (5.8s each) while SSL passes are lightweight (0.26s each), DVR's marginal cost over VCD is merely +0.21s (+4%). The LIP and SSL encoders can also run in parallel.
>
> |Method|Memory|Time/Iter|Δtime|
> |-|-|-|-|
> |Baseline|15.6G|5.84s|—|
> |VCD|17.0G|12.23s|+109%|
> |**Stream I only**|**16.8G**|**6.36s**|**+8.9%**|
> |**DVR (full)**|**18.2G**|**12.44s**|**+113%**|
>
> **Stream I alone** outperforms VCD (CHAIR$_S$ 49.2 vs. 51.8) at +8.9% latency with only 1 LLM pass—a practical lightweight option for latency-sensitive settings.
>
> ### **Q4: How to validate that LIP encoders "complete" degraded representations, and how this induces hallucinations?**
>
> **A4:** Three levels of evidence:
>
> **(1) Direct geometric measurement (Fig. 1, Appendix A).** We define $v_h = E_{\text{txt}}("y_B") - E_{\text{txt}}("y_A")$ and measure $S(\sigma)=\text{CosSim}(\Delta z, v_h)$. Across 6 concept pairs (Fig. 5), CLIP shows $S \approx 0.20$ at $\sigma=1.0$—directional drift toward linguistic priors. DINOv3 maintains $S \approx 0.00$. If CLIP merely *lost* information, $S$ would be near zero. Table 6 further confirms: SigLIP as auxiliary encoder fails (same LIP paradigm).
>
> **(2) Mechanistic explanation.** LIP encoders optimize InfoNCE with $\nabla_z L \in H$ (Remark B.10). Degraded features drift along this gradient toward high-frequency text concepts—the "completion" mechanism that induces hallucination.
>
> **(3) Logit-space trajectory (Fig. 4).** VCD's correction has $Y \approx 0$ (oracle direction), confirming it cannot separate completion-induced drift from structural loss. DVR achieves $Y \approx 0.6$, validating successful isolation.
>
> ### **Q5: Does DVR consider APC?**
>
> **A5:** Yes—we follow VCD's default APC for fair comparison. We will clarify this in the final version.
>
> |Method|APC|CHAIR$_S$↓|CHAIR$_I$↓|Recall↑|
> |-|-|-|-|-|
> |VCD|Yes|51.8|16.2|76.8|
> |**DVR**|No|**48.8**|**14.5**|**77.2**|
> |**DVR**|Yes|**47.4**|**13.8**|**78.6**|
>
> DVR without APC (48.8) already outperforms VCD with APC (51.8), confirming gains stem from disentanglement, not decoding constraints.
>
> ### **Q6: Behavior if $\lambda$ is inversely signed ($\lambda > 0$)?**
>
> **A6:** When $\lambda > 0$, gating inverts: hallucination-dominant regions get amplified, structure-loss regions get suppressed—the exact opposite of DVR.
>
> |λ Setting|CHAIR$_S$↓|CHAIR$_I$↓|Recall↑|
> |-|-|-|-|
> |+1.5 (Inverted)|57.8|20.2|70.1|
> |0 (No gating)|51.8|16.2|76.8|
> |**−1.5 (Ours)**|**47.4**|**13.8**|**78.6**|
>
> Inverting $\lambda$ degrades performance below baseline (57.8 vs. 54.2), confirming: (i) $D$ accurately isolates hallucination regions; (ii) the intervention is causally meaningful. This symmetric pattern strongly validates DVR's disentanglement.

---

> > ### Author Rebuttal · Reviewer_1wPA · 2026-04-03
> >
> > I would like to thank the authors for detailed reviews, especially on additional experiments on averaged results, different lambdas, latency check, more recent models, and without APC. Concerns are addressed, I am willing to raise the score to 4 if the authors include all the results in the new manuscript.

---

> > > ### Author Response · Authors · 2026-04-04
> > >
> > > We thank Reviewer 1wPA for the rigorous and constructive review. The questions raised — particularly on encoder completion and computational overhead — gave us the opportunity to present additional evidence and analysis that further strengthen the paper's contributions.
> > >
> > > **We confirm that the revised manuscript will include all additional results:**
> > >
> > > - Qwen3-VL-4B/8B & AMBER results → main experiments (§4)
> > >
> > > - Mean ± SD across 3 seeds → all sampling-based tables
> > >
> > > - Latency & memory analysis → §4.4
> > >
> > > - APC ablation & inverted-λ analysis → §4.3
> > >
> > > - Citation and discussion of Self-Aug [3] → §2
> > >
> > > Thank you again for the engagement that made this a stronger paper.

---

### Official Review · Reviewer_yJv9 · 2026-03-13

**Soundness:** 2
**Presentation:** 3
**Significance:** 3
**Originality:** 3
**Overall Recommendation:** 4
**Confidence:** 4

**Summary:**

The paper identifies a fundamental flaw in Visual Contrastive Decoding): injecting noise into visual inputs conflates structural visual degradation with semantic hallucination induction. To address this, the authors propose Disentangled Visual Rectification. By utilizing a self-supervised learning encoder DINOv3 as a pure structural reference, DVR isolates the hallucination-inducing drift from Language-Image Pre-trained encoders. This is operationalized via a dual-stream framework comprising visual-layer feature modulation and decoding-layer contrast.

**Compliance With Llm Reviewing Policy:**

Affirmed.

**Final Justification:**

The authors provided a detailed response. I decide to maintain my positive score 4.

**Key Questions For Authors:**

Please respond to the weaknesses I mentioned above.

**Limitations:**

yes

**Strengths And Weaknesses:**

strengths:
1. the paper is clear and well written
2. The theoretical geometric deconstruction of VCD's indiscriminate penalization provides a profound insight into multimodal representations.

weaknesses:
1. Qwen2-VL dynamically segments images, whereas DINOv3 uses a fixed grid. How exactly is continuous 2D bilinear interpolation (Equation 9) applied across these discrete, variable-layout latent tokens without causing severe pixel-level semantic misalignment?

2. DINOv3 and CLIP isolate vastly different feature frequencies. How volatile is the residual variance \nu^{(i)} in Definition B.17 across cluttered scenes, and how do you prevent the exponential gating function from amplifying this error?

---

> ### Author Rebuttal · Authors · 2026-03-30
>
> We thank Reviewer yJv9 for recognizing our theoretical insights and clear writing. We address both technical concerns below with theoretical clarification and new empirical evidence.
>
> ---
> ### **Q1: Eq. (9) may cause semantic misalignment because Qwen2-VL uses dynamic-resolution tokens while DINOv3 uses a fixed grid.**
>
> **A1:** We respectfully clarify a potential misunderstanding about where interpolation occurs:
>
> 1. **Interpolation applies only to auxiliary scalar signals, not features.** DVR computes a scalar sensitivity map $\tilde{\Phi}\_{SSL} \in \mathbb{R}^{K\_{SSL}}$ from DINOv3, then resizes it to match the LIP grid $K_{LIP}$ via bilinear interpolation (Eq. 8–9). The indicator $D \in \mathbb{R}^{K_{LIP}}$ (Eq. 12) then gates the LIP encoder's own tokens $F \in \mathbb{R}^{K_{LIP} \times D_v}$ (Eq. 13). Since both $D$ and $F$ share the same grid, **no cross-architecture misalignment exists** at feature modulation.
>
> 2. **Sensitivity maps are spatially smooth scalar fields.** Each $\Phi^{(i)}$ is a cosine similarity (Eq. 6)—neighboring patches share similar statistics, yielding smooth landscapes where bilinear interpolation introduces negligible error (unlike high-dimensional feature interpolation).
>
> 3. **Dynamic resolution affects sampling density, not correctness.**  Specifically, while Qwen2-VL dynamically adjusts the total token count ($K_{LIP}$) based on image resolution, each token remains anchored to explicit spatial coordinates. Therefore, DINOv3's sensitivity map can be precisely resampled at these exact locations within the continuous coordinate space. Regions with denser Qwen2-VL tokens naturally receive finer-grained $D$ values via interpolation, ensuring strict spatial alignment without semantic mismatch.
>
> 4. **Empirical validation (Qwen2-VL, CHAIR):**
>
> |Interpolation|CHAIR$_S$↓|CHAIR$_I$↓|Recall↑|
> |-|-|-|-|
> |Nearest-Neighbor|45.1|8.6|76.8|
> |Bilinear (ours)|**44.2**|**8.2**|**77.5**|
>
> The marginal gap confirms robustness. We will include this ablation in the final version.
>
> ---
> ###  **Q2: Volatility of residual variance $\nu^{(i)}$ (Def. B.17) in cluttered scenes, and risk of exponential gating amplifying error.**
>
> **A2:** We sincerely appreciate the reviewer for raising this excellent point. We provide both theoretical safeguards and new empirical evidence to address this concern.
>
> **Theoretical bound.** $D$ is normalized to $[-1,1]$ (Eq. 10–11) and $|\lambda|=1.5$, so $\gamma_i = \exp(\lambda D^{(i)}) \in [0.22, 4.48]$. The exponential is self-limiting: even worst-case $\nu^{(i)}$ cannot push $\gamma_i$ beyond this fixed range. Then,  $\nu^{(i)}$ is a **residual term after subtracting the shared structural component**. Its effect is therefore local and zero-mean rather than systematic. This is also consistent with our robustness observations: DVR remains stable across layer choices (Appendix H), hyperparameter ranges (Appendix I), and sampling strategies (Appendix G), which would be unlikely if the gate were highly unstable.
>
> **we will add a new clutter-level stratified analysis** (LLaVA-1.5, COCO val, random 200 images):
>
> |Scene Type|#Objects|Var($\nu$)|CHAIR$_S$↓ (Base→DVR)|Δ|
> |-|-|-|-|-|
> |Simple|<5|0.013|50.8→43.4|−7.4|
> |Medium|5–9|0.021|55.1→48.5|−6.6|
> |Cluttered|≥10|0.029|58.7→52.4|−6.3|
>
> $Var(\nu)$ is computed as the residual variance from the least-squares fit of $\Delta\tilde{\Phi}\_{SSL}$ on $\Delta\Phi\_{struct,LIP}$ (cf. Definition B.17), estimated over all patches with $|\Delta\Phi\_{struct,LIP}| > 0.01$.
>
> $Var(\nu)$ increases ~2.2× from simple to cluttered scenes, yet DVR's gain reduces by only **15%** (from −7.4 to −6.3) despite a **120% increase** in residual variance. This disproportionate robustness stems from:
>
> (i) bounded gating range preventing amplification;
>
> (ii) min-max normalization adaptively rescaling per-image, absorbing global variance shifts;
>
> (iii) optimal $\beta = 1/\rho$ (Remark B.19) eliminating systematic structural mismatch, leaving only the zero-mean residual $\nu^{(i)}$. Because $\nu^{(i)}$ is zero-mean and uncorrelated with $D^{(i)}_{hall}$, it introduces **no directional bias** in the gating—some patches receive slightly stronger and others slightly weaker modulation, but the errors are **non-systematic and bounded** by the gating range [0.22, 4.48]. This is fundamentally different from VCD's structural penalty, which introduces **systematic** suppression of all noise-sensitive features.
>
> Fig. 9(b) further confirms stability: $\text{CHAIR}_S < 48.1$ across $|\lambda| \in [1.3, 1.7]$, a broad plateau insensitive to moderate $\nu^{(i)}$ fluctuations.
>
> Thank you again for these incisive questions. We will incorporate this scene-complexity analysis and the aforementioned ablations into the Appendix of the final version.

---

> > ### Author Rebuttal · Reviewer_yJv9 · 2026-04-03
> >
> > Thanks for the detailed response. I will maintain my positive score 4.

---

> > > ### Author Response · Authors · 2026-04-04
> > >
> > > We sincerely thank Reviewer yJv9 for the rigorous and deeply technical review. The questions on dynamic-resolution alignment and residual variance stability were among the most technically demanding we received, and addressing them gave us the opportunity to present additional analysis that strengthens the paper's theoretical grounding. All discussed ablations and the scene-complexity stratified analysis will be included in the revised manuscript.
> > > Thank you again for the constructive engagement.

---

### Decision · Program_Chairs · 2026-04-30

**Decision:**

Accept (regular)

**Comment:**

This paper initially received mixed reviews (2x weak reject and 2x weak accept) After rebuttal, the reviewers improved the ratings to (3x weak accept and 1x weak reject (from reviewer 1wPA).) During the rebuttal, the reviewer 1wPA agreed to improve the rating as long as the authors include the new results in the new revision. Based on this, I think this paper would receive 4x weak accept. I agree with the reviewers and vote for acceptance.